# Specific and redundant roles for Gli2 and Gli3 in establishing cell fate during murine hair follicle development

Gokcen Gozum[1], Lakshit Sharma[1,7], Paula Henke[1,7], Lisa Wirtz [ID] [2,3,4], Mareike Damen[2,3], Viktoria Reckert[1], Peter Schettina[1], Melanie Nelles[1], Craig N Johnson [ID] [5], Hisham Bazzi[2,3,4,5] & Catherin Niemann [ID] [1,6 ✉]

## Abstract

**Formation of skin epithelial appendages like hair follicles requires hedgehog (Hh) signal reception, its conduction through the primary cilium and activation of Gli transcription factors. How Hh signalling induces cell-type-specific responses through Gli transcription factors in hair follicle stem cells and their cilia-dependence remains unclear. Here, we use conditional mouse mutants to genetically dissect the roles of Gli2 and Gli3 transcription factors and cilia in the skin epithelium. Upon keratinocyte-specific depletion of Gli2, hair follicle morphogenesis is delayed whereas sebaceous gland formation is enhanced, suggesting a dual role for Gli2 during appendage development. Gli2 promotes proliferation of sebaceous gland stem cells, impacting the number and size of individual sebaceous gland lobes. While ablation of Gli3 shows no detectable phenotypes, hair follicle cell fate is blocked in Gli2/Gli3 double knockout (dKO) mice, suggesting functional compensation. Finally, loss of cilia phenocopies the depletion of Gli2 but not the Gli2/3 dKO mutants. Our study reveals compartment-specific regulation of murine skin morpohogenesis by Gli2 and cilia-independent activator functions of Gli3 in the absence of Gli2.**

**Keywords** Hedgehog; Gli2; Hair Follicle; Sebaceous Gland; Stem Cell
**Subject Categories** Cell Adhesion, Polarity & Cytoskeleton; Development; Stem Cells & Regenerative Medicine

## Introduction

Deciphering the molecular and cellular networks driving the induction and maintenance of complex epithelial structures remains a major challenge in regenerative medicine. Morphogenesis of the mammalian skin involves an intricate crosstalk between different cell types and the hedgehog (HH) signalling pathway emerged as a key player not only for the development of epithelial appendages, like hair follicles (HF) (Chiang et al, 1998; Sennett and Rendl, 2012), but also for the process of HF neogenesis of injured skin (Lim et al, 2018; Sun et al, 2020; Frech et al, 2022; Liu et al, 2022). Importantly, abnormal activation of HH in epithelial cells induces cancer and constitutes the main driver for basal cell carcinoma (BCC) formation in the skin (Oro et al, 1997), thus highlighting the necessity to better understand the cellular mechanisms of HH signalling particularly in the skin epithelium, to allow for therapeutic strategies preventing disease formation.

The HH signal is translated into cell-specific responses by members of the Glioma-associated oncogene homologue (Gli) family of transcription factors (Zhang and Beachy 2023; Briscoe and Thérond, 2013). Upon HH ligand exposure, cells respond by posttranslational processing of the Gli protein into predominantly transcriptional activators (Gli$^A$). In the absence of HH ligands, Gli proteins can function as transcriptional repressors (Gli$^R$), preventing the expression of HH target genes (Aza-Blanc et al, 1997; Méthot and Basler, 1999; Niewiadomski et al, 2014). Mammalian HH signalling and Gli processing into Gli$^A$ or Gli$^R$ may require the primary cilium, an antenna-like protrusion present in most postmitotic cells in the mammalian body (Nozawa et al, 2013; Lee et al, 2016). The movement of HH signalling intermediates into and out of the cilium is facilitated by intraflagellar transport (IFT) proteins. Defects in various components of cilia result in severe birth abnormalities termed ciliopathies that are also characterised by misregulation of HH target genes (Huangfu et al, 2003; Ho and Stearns, 2021).

Mechanistic studies showed that in the absence of HH ligands, the HH receptor patched 1 (Ptch1) suppresses its co-receptor smoothened (Smo). HH ligand binding to Ptch1 inhibits Smo suppression, allowing it to move to the primary cilia and activate HH signal transduction, including Gli-mediated expression of HH target genes (Zhang and Beachy, 2023). The disruption of the IFT complexes, including IFT88, leads to the loss of cilia and subsequently interferes with HH signalling (Huangfu et al, 2003; Houde et al, 2006). The role of primary cilia in HH signal transduction is complex and context-dependent. Given the

[1]Center for Molecular Medicine Cologne (CMMC), University of Cologne, Cologne, Germany. [2]Department for Dermatology and Venerology, University Hospital Cologne, Cologne, Germany. [3]The Cologne Cluster of Excellence in Cellular Stress Responses in Aging-associated Diseases (CECAD), University of Cologne, Cologne, Germany. [4]Department of Cell Biology of the skin, University of Cologne, Medical Faculty, Cologne, Germany. [5]Cell & Developmental Biology, University of Michigan Medical School, Ann Arbor, MI, USA. [6]Center for Biochemistry, University of Cologne, University Hospital Cologne, Cologne, Germany. [7]These authors contributed equally: Lakshit Sharma, Paula Henke. ✉E-mail: cnieman1@uni-koeln.de

importance of the HH-cilia signalling axis in numerous developmental processes and malignancies, a better understanding of the function of cilia in the processing and activation of Gli proteins is crucial.

HH signalling is essential for epidermal appendage formation in mammalian skin. Previous studies in Shh$^{-/-}$ and Gli2$^{-/-}$ mice revealed that Shh signalling, although dispensable for HF initiation, is required for HF down-growth past the hair germ stage of appendage formation. In particular, epithelial proliferation is decreased in developing HFs in both Shh$^{-/-}$ and Gli2$^{-/-}$ mice (St-Jacques et al, 1998; Chiang et al, 1998; Mill et al, 2003). The expression of Gli1, one of the three transcription factors and also a target gene of active HH signalling, is regulated by epithelial Gli2 activity in the skin. However, the loss of Gli1 does not disrupt mouse development, suggesting that Gli1 does not play a major role in mammalian skin development (Park et al, 2000; Bai et al, 2002; Mill et al, 2003). Earlier results implicated Gli2 as the primary activator downstream of HH signalling and key transcription factor mediating HH responses in the epidermis (Mill et al, 2003). However, due to potential functional redundancy of Gli2 and Gli3, the specific roles of the individual transcription factors in the mammalian skin epithelium have not been reported.

During HF morphogenesis, the HH response is induced in both, the epithelium and stromal cells, indicating that HH signal reception occurs in different HF compartments. Mouse models with a dermis-specific loss of Smo function showed the loss of dermal papilla precursor cells (Woo et al, 2012). Disrupting cilia assembly in the ventral dermis leads to an early arrest of HF development, similar to the HF defect observed in Shh$^{-/-}$ and Gli2$^{-/-}$ mice (Lehman et al, 2009; St-Jacques et al, 1998; Chiang et al, 1998; Mill et al, 2003). The *Ift88* and *Kif3a* cilia mutant mice with small or absent dermal condensates revealed that dermal cilia are critical signalling components for normal HF morphogenesis (Lehman et al, 2009). More recently, it has been shown that HH activation in dermal papilla fibroblasts by expressing an active Smo mutant can induce new hair growth and increase fibroblasts heterogeneity (Liu et al, 2022). Genetic mouse experiments analysing IFT cilia mutants showed that the molecular trafficking machineries can be segregated according to their function in either ciliogenesis or in mediating HH signalling (Yang et al, 2015). However, how cilia function is linked to Gli processing and HH pathway activation in the mammalian skin epithelium still remains unclear.

Here, we use genetic approaches combined with single-cell transcriptome analysis to identify Gli2 as a crucial transcription factor executing context-specific functions in skin appendage formation. Upon depletion of Gli2, but not Gli3, from the murine epidermis, HF morphogenesis is delayed, whereas sebaceous gland (SG) formation is promoted, demonstrating a dual role for Gli2 in different HF sub-compartments. Mechanistically, our data reveal that Gli2 specifically regulates the proliferation of SG stem cells, thereby controlling the development and maintenance of the number and size of individual SG lobes. Ablation of both, Gli2 and Gli3 in the skin epithelium, suggests redundant functions for both transcription factors. HF morphogenesis and differentiation are blocked in double knockout mice, mimicking the skin phenotype observed upon epidermal depletion of the HH receptor Smo. Finally, our work reveals cilia-independent Gli activator functions of Gli3 in Gli2 epidermal knockout mice.

# Results

## Keratinocyte-specific ablation of Gli2, but not Gli3, causes defective hair follicle formation

To test the role of Gli transcription factors in the murine epidermis, we crossed Gli2$^{fl/fl}$ mice with K14Cre transgenic mice that express Cre recombinase from embryonic day (E) 10 onwards (Corrales et al, 2006; Hafner et al, 2004). At birth, K14Cre$^{tg}$;Gli2$^{fl/fl}$ mice (thereafter referred to as Gli2$^{EKO}$), lacking Gli2 specifically in keratinocytes, were macroscopically indistinguishable from Gli2$^{fl/fl}$ littermates. However, histological analysis of back skin tissue revealed that HF development was just initiated in Gli2$^{EKO}$ newborn mice at postnatal day 0 (P0), whereas HFs were already growing deep into the underlying dermal tissue in control mice, indicating a strong delay or block in HF morphogenesis in Gli2$^{EKO}$ animals (Fig. 1A). The hair growth defect became macroscopically visible with impaired hair coat formation in Gli2$^{EKO}$ mice at postnatal day 6 (P6) (Fig. EV1A). Analysis at P6 further demonstrated that the number of HFs was significantly reduced in Gli2$^{EKO}$ mice (Fig. 1B,C). Next, we investigated whether the lower HF number was due to a growth defect of a particular type of pelage hair. Compared to control littermates, Gli2$^{EKO}$ mice showed a dramatic decrease in zigzag hairs (third wave, ~75% of pelage hair) and a slight increase in auchene/awl type of HF (second wave, ~25% of pelage hair), while guard hairs (~2%, first wave of pelage hair) were not changed (Fig. 1D). Our data demonstrate that epidermal Gli2 activity is required for HFs forming in the second and third wave of pelage hair morphogenesis, starting at E16.5 and E18.5, respectively.

To understand the role of epidermal Gli3 in appendage formation, we also generated a conditional epidermis-specific knockout for Gli3 (thereafter Gli3$^{EKO}$ mice) (Blaess et al, 2008; Hafner et al, 2004). Gli3$^{EKO}$ mice could not be distinguished from their littermate controls shortly after birth and Gli3$^{EKO}$ skin histology appeared similar to that of control animals (Fig. 1A). In contrast to Gli2$^{EKO}$ mice, HF morphogenesis, including HF number, appeared normal in Gli3$^{EKO}$ animals (Fig. 1B,C), consistent with the phenotype seen in Gli3$^{-/-}$ mice (Mill et al, 2003). Together, the data indicate an important role for epidermal Gli2, but not Gli3, in the early stages of secondary and tertiary HF morphogenesis.

## Epithelial Gli2 and Gli3 have redundant functions in hair follicle morphogenesis

To address the possibility that Gli2 and Gli3 exert redundant functions and compensate for each other in the mouse epidermis, we generated Gli2/3$^{EKO}$ double knockout mice by depleting both transcription factors from the epidermal tissue. Unlike Gli2$^{EKO}$ and Gli3$^{EKO}$ single mutants, Gli2/3$^{EKO}$ double mutants had significantly decreased body weight and showed severe skin abnormalities that became macroscopically apparent at P6 (Fig. EV1A–E). At birth, Gli2/3$^{EKO}$ mice showed histologically abnormal and blocked hair growth (Fig. 1E). However, skin abnormalities were much more severe compared to control and Gli2$^{EKO}$ mice (Fig. 1A), where Gli2/3$^{EKO}$ mice did not show any hair growth (Fig. EV1F). The skin defects were particularly prominent starting at P6, a time when control littermates had fully developed HFs reaching into the skin fat layer, and Gli2/3$^{EKO}$ mice exhibited deformed epithelial

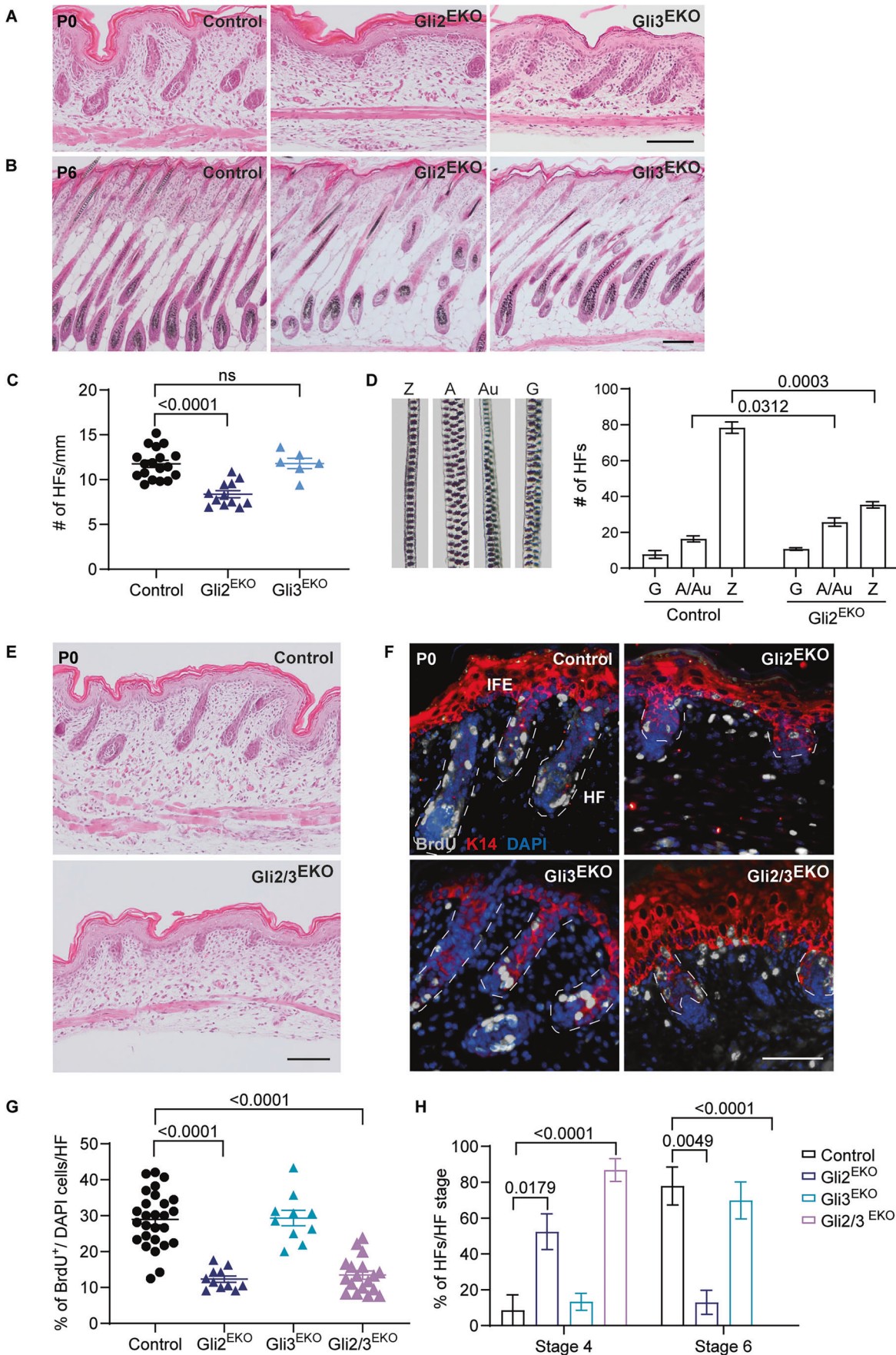

**Figure 1.   Synergistic functions of Gli2 and Gli3 are crucial for normal hair follicle morphogenesis.**

(A, B) Representative H&E staining of back skin sections from P0 (A) and P6 (B) Gli2[EKO], Gli3[EKO] and control littermates. (n = 3 mice/genotype). (C) Quantifications of HFs in back skin from P6 Gli2[EKO], Gli3[EKO] and control mice. Three different areas per animal were counted (n = 3–6 mice/genotype) (p value = 1.74e-06). (D) Representative images (left) of Zigzag (Z), Awl (A), Auchene (Au) and Guard (G) hair types and quantification (right) of 100 back skin HFs from P16 Gli2[EKO] and control mice. (n = 3 mice/genotype). (E) Representative H&E staining of back skin sections from P0 Gli2/3[EKO] and control littermates. (n = 3 mice/genotype). (F) Immunofluorescence staining for BrdU (grey), K14 (red) and DAPI (blue, nuclei) of back skin sections from P0 Gli2[EKO], Gli3[EKO], Gli2/3[EKO] and control mice. (n = 3–4 mice/genotype). Dashed lines mark HFs. (G) Quantification of BrdU+/DAPI+ ratio in back skin HFs in P0 Gli2[EKO], Gli3[EKO], Gli2/3[EKO] and control mice. About 2–6 HFs per animal were counted (n = 2–4 mice/genotype) (p value = 2.17e-10 for Gli2[EKO] and 2.23e-10 for Gli3[EKO]). (H) Quantification of the percentage of HFs/HF stages 4 and 6 in P0 Gli2[EKO], Gli3[EKO], Gli2/3[EKO] and control mice. About 4–10 HFs per animal were counted (n = 3–5 mice/genotype) (p value = 8.02e-05 for stage 4 and 8.13e-05 for stage 6 Gli2/3[EKO]). Scale bars 100 µm (A, B, E) and 50 µm (F). Data were presented as mean ± SEM. P value calculated using unpaired Student's t-test. Source data are available online for this figure.

invaginations and cyst-like structures that were not growing into the fat tissue of the skin (Fig. EV1F). These results demonstrate that epidermal Gli2 and Gli3 together are required for appendage formation in the mammalian skin. The data also point to redundant functions for Gli2 and Gli3 because single knockout mice showed milder (Gli2[EKO] mice) or no detectable (Gli3[EKO] mice) skin phenotypes.

Because HH signalling is associated with cell proliferation in the skin (St-Jacques et al, 1998; Chiang et al, 1998; Mill et al, 2003; Park et al, 2018), we next investigated whether changes in HF progenitor cell proliferation were associated with the HF defects in Gli2[EKO] and Gli2/3[EKO] mutant mice. Analysis of BrdU incorporation in newborn mice showed strong reduction of BrdU-positive cells in developing HFs of both, Gli2[EKO] and Gli2/3[EKO], but not Gli3[EKO] mice, when compared to control littermates (Fig. 1F,G). The decrease in proliferation was also reflected by the differences in HF morphogenesis: whereas the majority of HF of Gli3[EKO] and control mice where mainly in stage 6 of HF development, follicles of Gli2[EKO] and Gli2/3[EKO] mice were still delayed at stage 4 (Fig. 1H) (Paus et al, 1999; Saxena et al, 2018).

The contribution of both Gli2 and Gli3 to normal appendage formation in mammalian skin prompted us to investigate whether these two transcription factors are mediating the collective HH signalling response in the mammalian epidermis. We thus asked how the phenotypes of the individual and double Gli knockout mice compare to the skin phenotype of mice with the epidermal deletion of Smo, the membrane receptor for HH signalling (thereafter Smo[EKO] mice). Smo[EKO] mice were generated by crossing Smo[fl/fl] line with the K14Cre mouse line (Long et al, 2001; Hafner et al, 2004). Importantly, the skin phenotype of Smo[EKO] newborn mice looked very similar to Gli2/3[EKO] mice with an arrest in HF formation at the hair germ stage of morphogenesis (Fig. EV1G). Our observations in Smo[EKO] mice are supported by previous reports showing that depletion of Smo from mouse epidermis leads to defects in HF formation (Lichtenberger et al, 2016; Gritli-Linde et al, 2007).

## Gli2 and Gli3 are required for hair lineage differentiation and hair follicle stem cells

Next, we investigated whether Gli transcription factors play a role in HF stem and progenitor lineage differentiation and analysed the expression of specific hair differentiation markers at P6, when HF progenitors undergo differentiation. Immunofluorescence analysis detected keratin (K) 75, a marker for the HF companion layer, and the HF inner root sheath marker K71 in control and Gli3[EKO] mice.

The hair cortex marker K86 showed less expression during these early stages of HF formation and was normally expressed in Gli3[EKO] and littermate control mice (Fig. EV2A), demonstrating that hair lineages were normally generated in Gli3[EKO] mice. We also analysed P6 Gli2[EKO] mice, when the growth of some HF was regained after the initial block in HF morphogenesis seen in newborn mice (Fig. 1A–D). Indeed, immune stainings for hair keratins revealed that fewer HF expressed these hair lineage markers (Fig. EV2A). However, RNA analysis showed that K75, K71 and K86 expression were significantly reduced in Gli2[EKO] mice compared to control and Gli3[EKO] animals (Fig. EV2B,C), most likely reflecting the delay in HF morphogenesis and a reduction in HF number observed in Gli2[EKO] mice (Fig. 1A–C).

Remarkably, expression of hair lineage markers was completely blocked at P6 in Gli2/3[EKO] mice when compared to control skin samples (Fig. 2A). A block in hair differentiation marker expression was also detected at the mRNA level (Fig. 2B), indicating that HF progenitor cells do not undergo hair lineage differentiation in Gli2/3[EKO] mice. We then asked what epithelial cell types populate the abnormal HFs and cyst-like structures observed in Gli2/3[EKO] mutants. Interestingly, K10 and filaggrin, markers for supra-basal keratinocytes within the interfollicular epidermis, were both detected in the cyst-like abnormal HF structures in Gli2/3[EKO] mice, but not in HFs of littermate controls (Fig. 2C,F). Quantification of K10 and filaggrin protein on tissue sections showed a significant increase in abnormal HFs (Fig. 2D,G). Further, mRNA expression of Krt10 and filaggrin is strongly elevated in Gli2/3[EKO] epidermal tissue (Fig. 2E,H). Large cyst-like structures are maintained in adult Gli2/3[EKO] skin and are also positive for squamous marker expression, e.g. filaggrin (Fig. EV3A–F). These results indicate that both transcription factors, Gli2 and Gli3, together are required for HF progenitor lineage differentiation.

Given that hair differentiation was completely blocked in Gli2/3[EKO] mice, we wondered whether HF SCs are generated in Gli2/3[EKO] mice. To address this question, HF bulge SC marker K15 and CD34 were analysed. As expected, K15 is strongly expressed in growing HF at P6 and by bulge SCs localising to the lower permanent part of the telogen HF in adult control animals. However, K15 protein is not detected in abnormal cyst-like epithelial structures in Gli2/3[EKO] P6 and adult mice (Figs. 2I and EV3G) and marker quantification shows a significant reduction of K15 protein on skin sections (Figs. 2J and EV3H). However, Krt15 mRNA expression is not significantly changed in P6 and P49 epidermis of Gli2/3[EKO] mice (Figs. 2K and EV3I). Further, analysis of bulge SC marker CD34 revealed a strong reduction of CD34 protein in the skin epithelium of adult Gli2/3[EKO] mice (Fig. EV3J,K), whereas CD34 mRNA

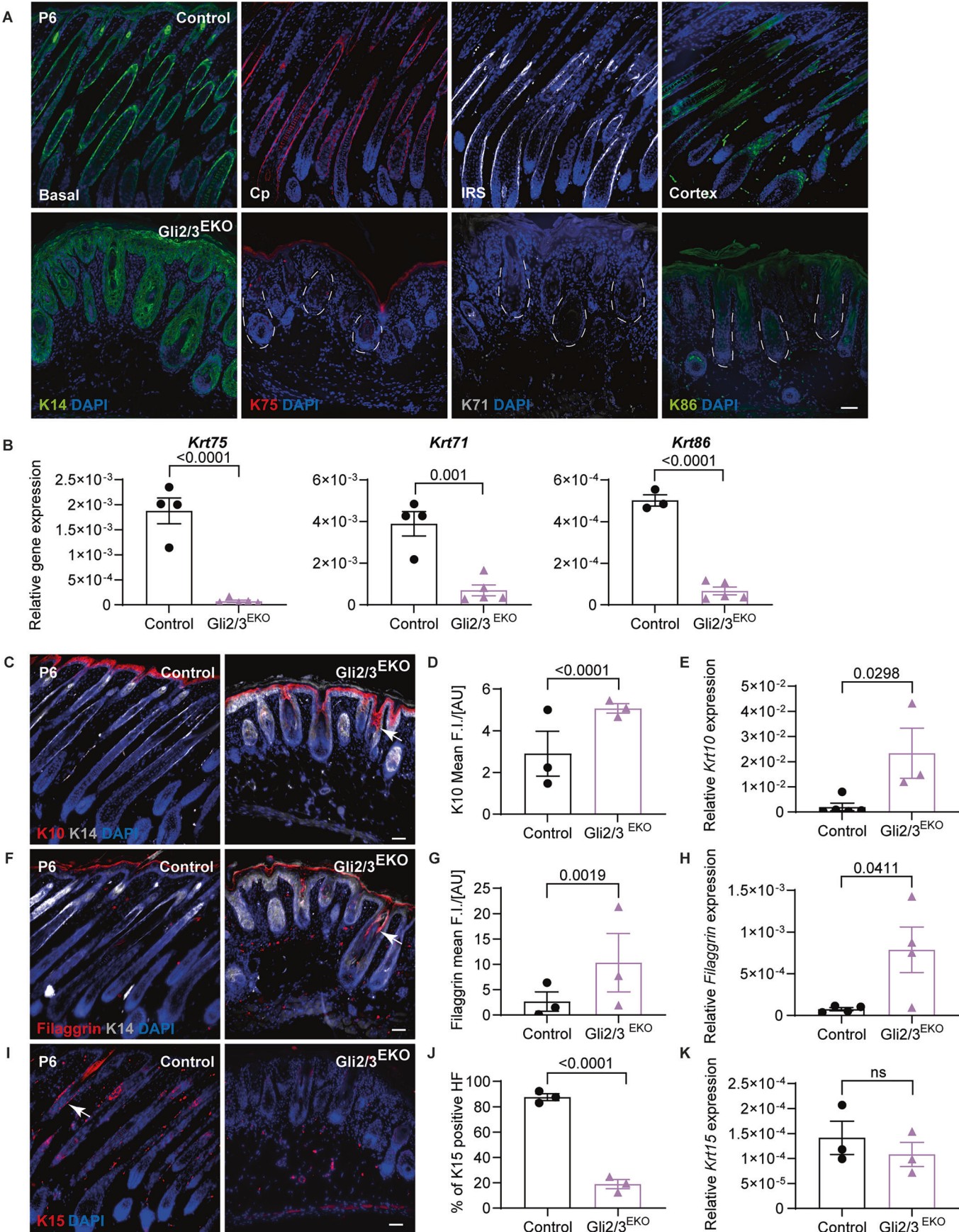

**Figure 2. Gli2 and Gli3 are required for hair lineage differentiation and hair follicle stem cells.**

(A) Immunofluorescence staining for K14 (green), K75 (red), K71 (grey), K86 (green) and DAPI (blue, nuclei) of back skin sections from P6 Gli2/3[EKO] and control littermates. (n = 3 mice/genotype). Dashed lines mark abnormal HFs in Gli2/3[EKO] skin. (B) qRT-PCR analysis for Krt75, Krt71 and Krt86 mRNA expression in back skin from P6 Gli2/3[EKO] and control littermates. A representative result from two technical replicates is shown. Each datapoint represents one animal (n = 3–5 mice/genotype) (p value = 9.66e-05 for Krt75 and 9.58e-06 for Krt86). (C, D) Immunofluorescence staining for K10 (red, arrow), K14 (grey) and DAPI (blue, nuclei) (C) and quantification of K10 (D) in back skin sections from P6 Gli2/3[EKO] and control littermates. (n = 3 mice/genotype). K10 fluorescence intensity (FI) per arbitrary unit (AU) (p value = 4.91e-05). (E) qRT-PCR analysis for Krt10 mRNA expression in back skin from P6 Gli2/3[EKO] and control littermates. Each datapoint represents one animal. (n = 3–5 mice/genotype). (F, G) Immunofluorescence staining for Filaggrin (red, arrow), K14 (grey) and DAPI (blue, nuclei) (F) and quantification of Filaggrin (G) in back skin sections from P6 Gli2/3[EKO] and control littermates. (n = 3 mice/genotype). Filaggrin fluorescence intensity (FI) per arbitrary unit (AU). (H) qRT-PCR analysis for Filaggrin mRNA expression in back skin from P6 Gli2/3[EKO] and control littermates. Each datapoint represents one animal (n = 4 mice/genotype). (I, J) Immunofluorescence staining for K15 (red, arrow) and DAPI (blue, nuclei) (I) and quantification of K15 (J) in back skin sections from P6 Gli2/3[EKO] and control mice. Each datapoint represents one animal, and 104–270 HFs per animal were counted. (n = 3 mice/genotype) (p value = 5.67e-16). (K) qRT-PCR analysis for Krt15 mRNA expression in back skin from P6 Gli2/3[EKO] and control littermates. Each datapoint represents one animal (n = 3 mice/genotype). Scale bars 50 µm (A, C, F, I). Data were presented as mean ± SEM. P value was calculated using an unpaired Student's t-test. Source data are available online for this figure.

expression was not significantly changed (Fig. EV3L). Together, these results demonstrate that Gli2 and Gli3 together are critical for the establishment and maintenance of bulge SCs and the differentiation of progenitor cells in distinct HF cell types.

## Gli2, but not Gli3 signalling, is cilia-dependent in the skin epithelium

Given that cilia play an important role in conducting the HH signal, we asked what roles do cilia play in mediating epithelial Gli transcription factor signalling in skin appendage formation? A cilia mutant Ift88[EKO] mouse line was analysed, deleting Ift88 from the skin epithelium by crossing Ift88[fl/fl] with K14Cre animals (Haycraft et al, 2007; Hafner et al, 2004; Damen et al, 2021). Importantly, Ift88[EKO] mice do not form cilia in the skin epithelium (Damen et al, 2021; Croyle et al, 2011). Initial HF formation was delayed in Ift88[EKO] newborn mice compared to control littermates (Fig. 3A). Later at P6, HF developed further and looked similar to those of control animals. However, the number of HFs was significantly reduced in Ift88[EKO] mice (Fig. 3B). The phenotype seen in newborn and young Ift88[EKO] mice appeared highly similar to the HF phenotype observed in Gli2[EKO] animals (Fig. 1A–C).

## Epidermal cilia-Gli2 signalling axis plays a specific role in the formation and maintenance of sebaceous glands

The resemblance of the skin phenotype seen in Gli2[EKO] and Ift88[EKO] mice also extended to the formation of sebaceous glands (SG), lipid-producing skin appendages normally attached to the HFs. At P6, SGs visualised by adipophilin staining appeared larger in the back skin of both Gli2[EKO] and Ift88[EKO] mice when compared to the littermate controls (Fig. 3C). These data show that, in contrast to HF development, the formation of SGs was not delayed but rather accelerated in Gli2- and cilia-defective mice, pointing to a cell-type-specific and context-dependent effect of HH signalling.

We wondered whether this is a transient defect during mouse development or that Gli2 signalling plays a general role in SG renewal and maintenance. To address this issue, we examined epidermal whole mounts of tail skin in adult mice, where two prominent sebaceous glands are attached to one HF and are easily accessible for more detailed characterisation. Our analyses revealed that the Nile Red-positive SGs were clearly enlarged in 7-week-old Gli2[EKO] mice when compared to control littermates (Fig. 3D). In

contrast, Gli3[EKO] mice displayed SGs of normal size (Fig. 3D). Next, we wanted to know whether SGs were also affected in adult cilia mutant mice. Interestingly, SGs of adult Ift88[EKO] mice were also enlarged and resembled the phenotype observed in Gli2[EKO] mice of the same age (Fig. 3D).

Quantitative analysis revealed that the majority of SGs attached to one HF in Gli2[EKO] and Ift88[EKO] mice consisted of six to ten individual lobes, whereas glands in control littermates consistently contained two lobes. Some HFs in both mutant mouse lines contained SGs made of ten or more lobes (Fig. 3E). The individual SG lobes were also significantly smaller in Gli2[EKO] and Ift88[EKO] mice (Fig. 3F). Consequently, due to the dramatic increase in the number of lobes, the overall volume of the SGs attached to one HF was evidently increased in Gli2[EKO] and Ift88[EKO] mice when compared to littermate controls (Fig. 3G). Together, the similar phenotype seen in Gli2[EKO] and Ift88[EKO] mice points to cilia-dependent signalling of Gli2 in SG morphogenesis and maintenance (Croyle et al, 2011).

## Gli2 signalling regulates sebaceous gland progenitor cells during appendage formation

Next, we investigated the potential role of Gli2 during SG morphogenesis using epidermal tail whole mounts. We focused on P6, when SG formation is initiated and the first mature sebocytes emerge within the upper part of the HF (Frances and Niemann, 2012). Staining for the sebocyte marker adipophilin revealed that the number of developing lobes were increased in Gli2[EKO] and Ift88[EKO] mice early during appendage formation (Fig. 4A), supporting our finding at P49 (Fig. 3D,E). The developing lobes are localised to the same region of the HF as seen in control mice. Quantifications showed that the size of the individual lobes and the entire SGs were significantly increased to more than twice as big in Gli2[EKO] and Ift88[EKO] mice compared to littermate controls (Fig. 4B). Notably, the size and shape of the developing HFs were abnormal in whole mounts from Gli2[EKO] tail epidermis. HFs were significantly thicker and shorter compared to controls (Fig. 4C,D), indicating that the proper coordination of growth within the HF structures was abnormal in P3 and P6 Gli2[EKO] mice.

Furthermore, we wanted to find out whether Gli2 signalling controls the progenitor compartment that gives rise to mature sebocytes and SGs. Sebocytes have previously been shown to originate from Lrig1-positive (+ve) keratinocytes during SG

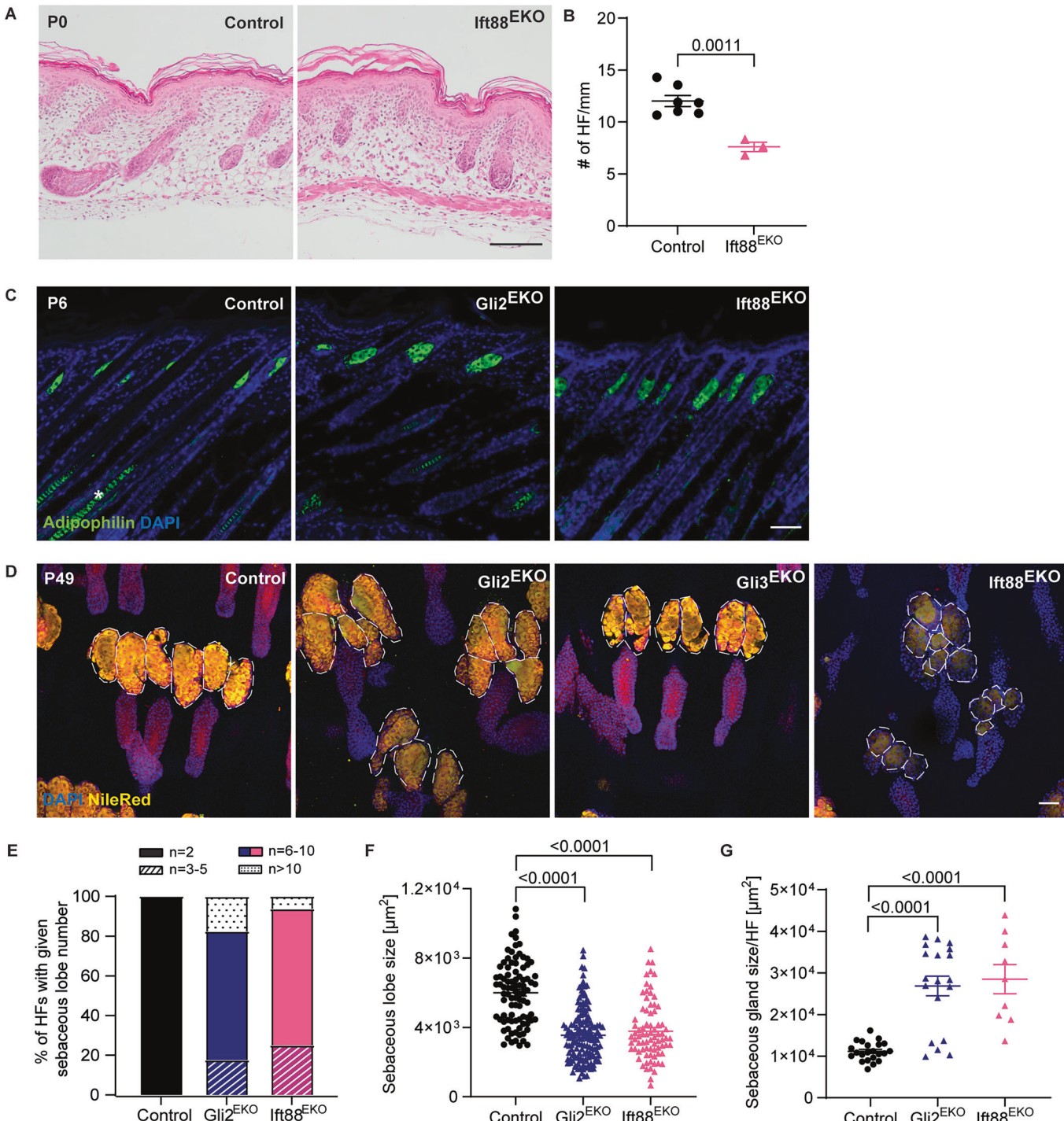

**Figure 3. Gli2, but not Gli3, signalling is cilia-dependent in the skin epithelium.**

(A) Representative H&E staining of back skin sections from P0 Ift88^EKO and control littermates. ($n = 3$ mice/genotype). (B) Quantification of HFs/mm back skin from P6 Ift88^EKO and control littermates. ($n = 3$–7 mice/genotype). (C) Immunofluorescence staining for Adipophilin (green) and DAPI (blue, nuclei) of back skin sections from P6 Gli2^EKO, Ift88^EKO and control mice. ($n = 3$ mice/genotype). (D) Nile red staining and DAPI (blue, nuclei) of epidermal whole mounts from tail skin from P49 Gli2^EKO, Gli3^EKO, Ift88^EKO and control mice. ($n = 3$ mice/genotype). Dashed lines mark the individual lobes of sebaceous glands. (E) Quantification of the percentages of HFs with a given number of sebaceous lobes in tail skin from P49 Gli2^EKO, Gli3^EKO, Ift88^EKO and control mice. 3–10 central HFs per animal were counted. ($n = 3$–4 mice/genotype).

(F, G) Quantification of the size of individual sebaceous lobes (F) and total SG size (G) in tail skin from P49 Gli2^EKO, Gli3^EKO, Ift88^EKO and control mice. 3–10 central HFs were counted ($n = 3$–4 mice/genotype) ($p$ value $= 2.31\text{e-}23$ for Gli2^EKO and $3.72\text{e-}14$ for Ift88^EKO in (F)) ($p$ value $= 2.02\text{e-}08$ for Gli2^EKO and $5.23\text{e-}08$ for Ift88^EKO in (G)). Scale bars 100 μm (A), 50 μm (C, D). Data were presented as mean ± SEM. Non-specific fluorescence signals are marked with an asterisk. $P$ value was calculated using an unpaired Student's $t$-test. Source data are available online for this figure.

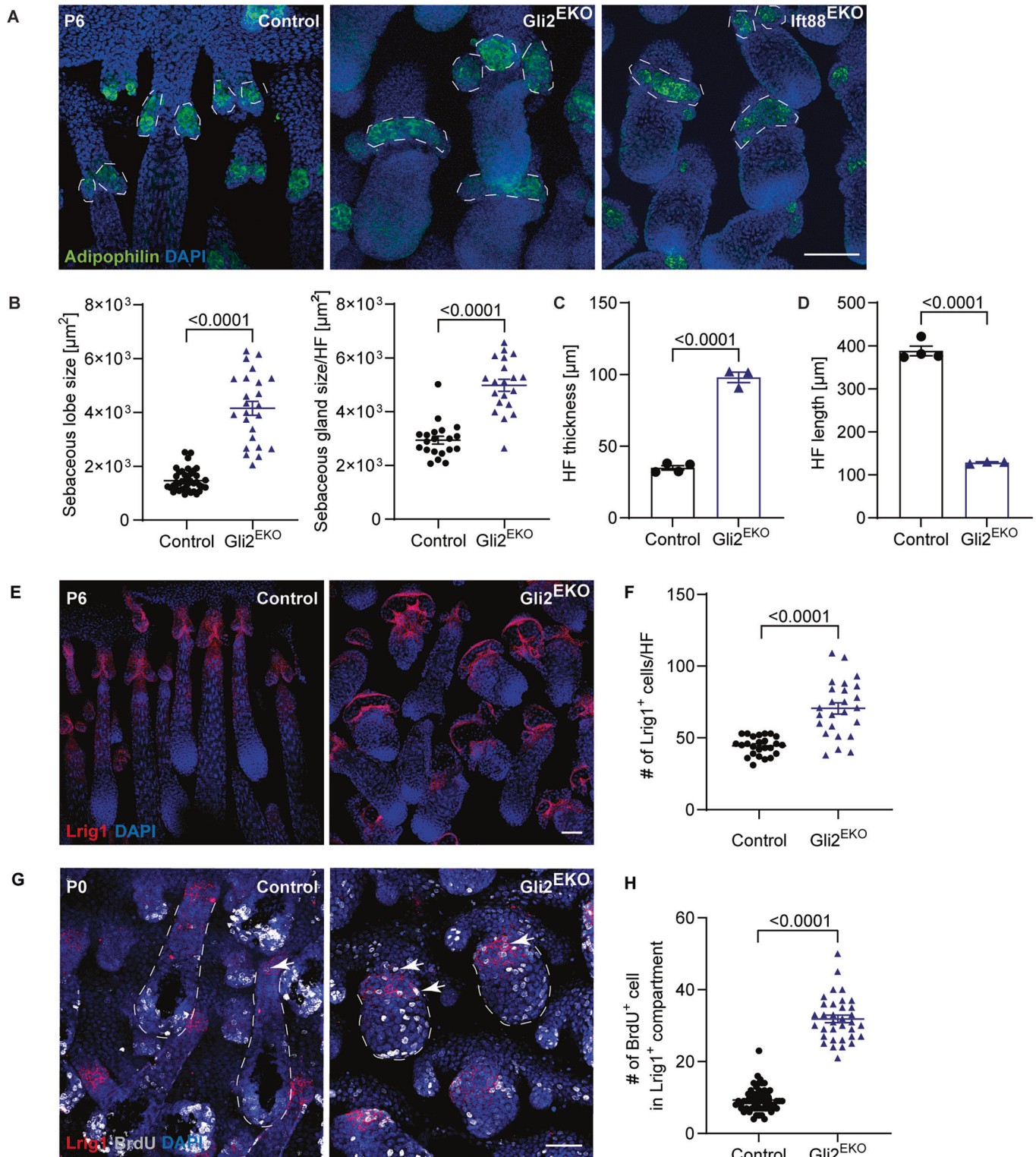

morphogenesis; we thus analysed epidermal whole mounts for the expression of Lrig1 (Frances and Niemann, 2012; Andersen et al, 2019). As expected, Lrig1 was strongly expressed in the junctional zone, adjacent to and surrounding the base of the two sebocyte clusters forming in the upper part of developing HFs at P6 control

mice (Fig. 4E). In contrast, the Lrig1+ve cell population was dramatically increased and expanded within the upper region of HFs, likely giving rise to the multiple lobes of the SGs in Gli2$^{EKO}$ mice (Fig. 4E). Quantitative analysis revealed a significant increase in the number of Lrig1+ve cells localising to the junctional zone of

**Figure 4.  Gli2 regulates sebaceous gland progenitor cells during appendage formation.**

(A) Immunofluorescence staining for Adipophilin (green) and DAPI (blue, nuclei) of tail skin from P6 Gli2$^{EKO}$, Ift88$^{EKO}$ and control mice. ($n = 3$ mice/genotype). Dashed lines mark sebaceous lobes/glands. (B) Quantification of the size of the individual sebaceous lobes (left) and total sebaceous gland (SG) (right) per HF in tail skin from P6 Gli2$^{EKO}$ and control littermates. Each datapoint represents the size of the individual lobe (left) or the total SG (right). 6–10 central HFs per animal were counted. ($n = 3$–4 mice/genotype) ($p$ value = 3.44e-21 for seb. lobes and 1.5e-06 for SG size). (C, D) Quantification of the thickness (C) and length (D) of HFs in epidermal whole mounts from tail skin from P3 Gli2$^{EKO}$ and control littermates. 7–16 central HFs per animal were counted. ($n = 3$–4 mice/genotype) ($p$ value = 1.05e-05 in (C) and 7.12e-06 in (D)). (E) Immunofluorescence staining for Lrig1 (red) and DAPI (blue, nuclei) in epidermal whole mounts of tail skin from P6 Gli2$^{EKO}$ and control littermates. ($n = 3$–4 mice/genotype). (F) Quantification of the number of Lrig1+ cells per HF. 7–10 HFs per animal were counted. ($n = 3$ mice/genotype) ($p$ value = 6.12e-08). (G, H) Immunofluorescence staining for BrdU (grey), Lrig1 (red) and DAPI (blue, nuclei) (G) and quantification of the number of BrdU+ cells in the Lrig1-compartment (H) in epidermal whole mounts of tail skin from P0 Gli2$^{EKO}$ and control littermates. Each datapoint represents one HF. 12–22 central HFs per animal were analysed. ($n = 2$–3 mice/genotype). Dashed lines mark the HFs and arrows mark BrdU/Lrig1 double-positive cells (G) ($p$ value = 6.64e-41). Scale bars 50 μm (A, E, G). Data were presented as mean ± SEM. $P$ value was calculated using an unpaired Student's $t$-test. Source data are available online for this figure.

the HF at P6 in Gli2$^{EKO}$ mice when compared to control animals (Fig. 4F). These data demonstrate that epithelial Gli signalling is required for normal SG development by restricting the size and localisation of the Lrig1+ve progenitor compartment during skin appendage formation.

Next, we addressed whether the establishment of the Lrig1-compartment is affected in epidermal Gli2$^{EKO}$ mice. Therefore, we analysed early HF morphogenesis at P0 and investigated cell proliferation in the absence of epidermal Gli2. BrdU incorporation experiments revealed that proliferation takes place in HF cells below the Lrig1+ve compartment in Gli2$^{EKO}$ and littermate controls. Remarkably, a significant increase in the number of BrdU+ve cells was detected within the Lrig1-compartment in Gli2$^{EKO}$ when compared to control mice (Fig. 4G,H; arrows in 4G), suggesting that Gli2 regulates SG formation by controlling the proliferation of Lrig1+ve progenitor cells.

## The cellular and molecular landscape of the Gli2$^{EKO}$ skin epithelium

To gain more insights into the full repertoire of cell types and gene expression programmes controlled by Gli2 signalling in early HF morphogenesis, we enzymatically dissociated the dorsal skin epidermis and HFs at P2, isolated keratinocytes from control and Gli2$^{EKO}$ mice, and generated single-cell transcriptome libraries. The full dataset consisted of 7199 cells, of which 5291 cells were keratinocytes (73.5%). Focusing on the keratinocytes, we used uniform manifold approximation and projection (UMAP) to explore the data layout in 2D. Applying graph-based Seurat clustering, we identified 16 clusters (Fig. 5A). The assignment of keratinocyte clusters revealed two main identities, e.g. IFE and HF cells, including a continuum of differentiation into distinct cell types/states in each (Fig. 5A).

Next, we used signature gene expression analyses to predict and assign cell-type identities as follows (Fig. 5B, with additional markers in Fig. EV4): (i) Keratinocytes in cluster 0, 7, 13 and 14 group together and show characteristics for matrix/progenitors (cluster 0), matrix cells expressing early HF lineage marker (cluster 13), inner layers of the IRS (cluster 7) and hair shaft (cluster 14). (ii) Clusters with strong expression of HF and ORS marker grouped in HF1 and HF2, with HF1 group encompassing clusters 1, 5 and 11 showing strong ORS marker expression and clusters 3, 6, 8 and 9 grouped in HF2, also showing ORS characteristics with additional marker expression, e.g. cluster 6 was enriched for HF stem cell marker, such as Lhx2, Sox9 and Nfatc1. Moreover, cluster 9 cells

presented characteristics of supra-basal hair keratinocytes, and cluster 8 contained a subpopulation of cells expressing marker molecules for early SG differentiation, including Plin2, Pparg and Krt79. (iii) Finally, keratinocytes expressing IFE markers group in either basal keratinocytes (clusters 4, 10 and 12) or supra-basal and differentiating keratinocytes in clusters 2 and 15 (Figs. 5B and EV4). Notably, given that the HF undergoes early stages of morphogenesis and cell differentiation at P2, cell identities did not appear well defined yet. For instance, many clusters contained keratinocytes co-expressing markers characteristic of different cell fates, when compared to published data in more developed skin (Rezza et al, 2016; Joost et al, 2020).

Next, we focused on the expression of HH signalling pathway components in control and Gli2$^{EKO}$ keratinocyte clusters. Ptch1 and Gli1, target genes of HH signalling, were preferentially expressed in cluster 0, encompassing matrix and HF progenitor cells, and in HF1-ORS cells of cluster 11 and 5 (Fig. 5C). Shh expression is only detected in matrix and hair progenitor cells of cluster 0, and the HH receptor Smo was enriched in HF matrix cells of clusters 0 and 13 and in HF1-ORS cells in cluster 11 (Fig. EV5A). Both, Gli2 and Gli3 were expressed by cells of the ORS lineages and keratinocytes of the hair matrix and hair lineages, and Gli3 was highly expressed by keratinocytes in states of early hair differentiation (Fig. EV5A). The overlap in Gli2 and Gli3 expression in HF keratinocytes supports our hypothesis that Gli3 compensates for Gli2 in Gli2$^{EKO}$ mice. The feature plots generally revealed that the HH component were expressed by fewer Gli2$^{EKO}$ keratinocytes compared to control (Figs. 5C and EV5A).

Based on our previous observations demonstrating a delay in HF growth in Gli2$^{EKO}$, we investigated whether the respective HF subpopulations showed smaller proportions in Gli2$^{EKO}$ skin compared to the control sample. As expected, hair matrix and inner HF layers in Gli2$^{EKO}$ mice showed a lower percentage of total cells (4.93, 1.51, 2.05 and 8.25% for clusters 7, 13, 14 and 0, respectively) compared to control cells (7.38, 3.48, 3 and 15.38%) (Fig. EV5B).

Next, we investigated whether HF differentiation is impaired in the dataset by analysing marker expression comparing Gli2$^{EKO}$ with control keratinocytes (Fig. 5D). Subtle reduction in expression of Myb, Msx1, Msx2 and Krt27 was observed in Gli2$^{EKO}$ HF matrix/progenitor cells (cluster 0). Marker expression for early IRS differentiation (e.g. Krt28, Krt25 and Krt71) in cluster 7 were similar and early HS differentiation (cluster 14) was not noticeably changed between Gli2$^{EKO}$ and control cells (Fig. 5D). Matrix cells that differentiate into HF lineages (cluster 13) show a small

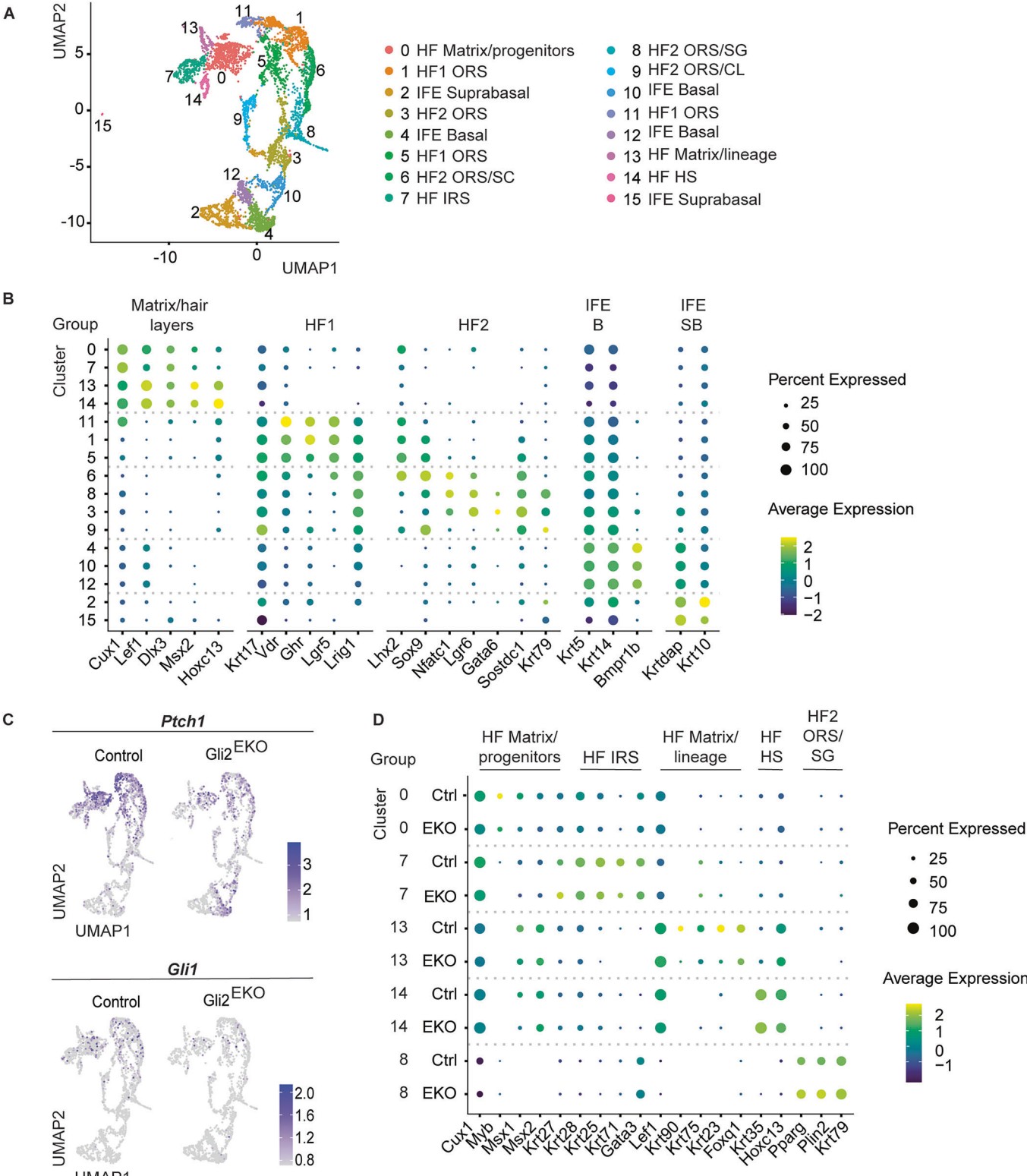

Figure 5. Molecular and cellular landscape of the Gli2$^{EKO}$ skin epithelium.

(A) UMAP embedding of P2 mice keratinocytes (left). Unsupervised Seurat-based clustering identified 16 clusters. List of all cell types identified (right). (B) Dotplot showing selected marker expression for each cluster. Colours represent scaled expression by gene, with dot size showing the percentage of cells in the cluster that express the gene. (C) Feature plots of scRNAseq data showing expression of Ptch1 and Gli1 in control and Gli2$^{EKO}$. Expression levels are colour coded, and expression is shown for values greater than 0.5. (D) Dotplot comparing marker expression between control vs Gli2$^{EKO}$ in different hair lineage cell clusters. Colours represent scaled expression by gene, with dot size showing the percentage of cells in the cluster that express the gene. ORS outer root sheath, SC stem cells, IRS inner root sheath, SG sebaceous gland, CL companion layer, HS hair shaft, B basal and SB supra-basal. Source data are available online for this figure.

reduction in Krt90, Krt75, Krt23 and Foxq1 expression in Gli2$^{EKO}$ samples, suggesting a delay in hair lineage differentiation (Fig. 5D). To gain more insights into the biological processes that are controlled by Gli2 signalling, we analysed differentially regulated genes in cluster 13 by performing a pathway and process enrichment analysis (Fig. EV5C). Interestingly, cellular processes required for tissue growth, molecule biosynthesis, cellular rearrangement (Rho signalling and cytoskeletal organisation) and melanosome assembly were all decreased in Gli2$^{EKO}$ keratinocytes compared to controls (Fig. EV5C). Together, these data point to an abnormal regulation of biological functions driving HF formation and cell specification in Gli2$^{EKO}$ mice.

Based on our initial observation, showing that SG differentiation was promoted in Gli2$^{EKO}$ mice (Fig. 3), we investigated marker expression for early sebocyte differentiation, that was detected in cluster 8. Remarkably, although cluster 8 of Gli2$^{EKO}$ mice contained slightly fewer cells (5.16%) compared to control mice (6.9%), expression of the sebocyte and SG lineage markers Plin2, Pparg and Krt79 were elevated in Gli2$^{EKO}$ cells (Fig. 5D). Moreover, the pathway and process enrichment analysis of differentially expressed genes in cluster 8 demonstrated an increase in biological processes involved in cell and lipid metabolism in Gli2$^{EKO}$ mice (Fig. EV5D). Taken together, the single-cell transcriptomic analysis demonstrated that, in contrast to the defective and delayed processes of HF lineage specification, SG cell differentiation is promoted in Gli2$^{EKO}$ mice, suggesting a dual role of Gli2 in skin appendage formation and confirming our findings using histology and immunofluorescence staining.

### Gli3 activator function in epidermal appendage formation

We finally wanted to gain more mechanistic insights into the redundant functions of Gli2 and Gli3 in the skin epidermis and aimed to better understand how Gli3 exerts compensatory activities in Gli2$^{EKO}$ mice. To address this, we tested the posttranslational processing of full-length Gli3 protein (Gli3$^{FL}$) into the Gli3 repressor (Gli3$^{R}$). Western blot analysis of P3 control epidermis revealed a strong Gli3$^{R}$ band (83 kDa) and a weaker Gli3$^{FL}$ band (190 kDa) that were both absent in epidermal samples from Gli3$^{EKO}$ mice (Fig. 6A), demonstrating the specificity of the antibody and detected bands. Interestingly, Gli3$^{R}$ protein appeared reduced in the isolated epidermis from Gli2$^{EKO}$ compared to control littermates (Fig. 6A). Protein quantification and determining the ratio between Gli3$^{FL}$ and Gli3$^{R}$ revealed a significantly higher proportion of Gli3$^{FL}$ relative to Gli3$^{R}$ in Gli2$^{EKO}$ epidermis, indicating that Gli3 might function as a transcriptional activator in Gli2$^{EKO}$ mice (Fig. 6A,B). This finding was supported by the analysis of the HH target gene Gli1. Whereas Gli1 mRNA levels did not significantly differ in Gli2$^{EKO}$ and Gli3$^{EKO}$ mice compared to control littermates, Gli1

expression was strongly reduced in Gli2/3$^{EKO}$ mice. These data show that HH signalling was still active in Gli2$^{EKO}$ mice and demonstrated that only the deletion of Gli2 and Gli3 together resulted in the total block of HH signalling in the skin epithelium (Fig. 6C).

## Discussion

A question of fundamental importance is how epithelial appendages are generated and organised into different cell types and functional tissue compartments (Abe and Tanaka 2017). Using a variety of different genetic knock-out mouse models and single-cell transcriptomics, this study dissects HH-driven molecular and cellular mechanisms underlying epidermal appendage formation. To our knowledge, our study shows for the first time a dual and context-dependent role for the transcription factor Gli2 during the morphogenesis of the pilo-sebaceous unit in the mammalian skin epithelium. On one hand, epidermal Gli2 is required for the proliferation of HF progenitors, whereas on the other hand, Gli2 restricts the expansion of SG progenitor cells. Based on our observation that early defects in HF formation seen in Gli2$^{EKO}$ and Gli2/3$^{EKO}$ newborn mice appeared very similar, the data suggest that Gli2 drives progenitor proliferation in HF keratinocytes at the early stages of HF morphogenesis (Fig. 6D,E). Furthermore, our results reveal that the initial block in HF formation in Gli2$^{EKO}$ mice is likely rescued by Gli3 at later stages of morphogenesis, demonstrating compensatory functions in the skin epithelium. Thus, our work identifies an important role for epidermal Gli3, a HH transcription factor that has previously been neglected in skin research, in skin appendage formation, specifically in the absence of Gli2.

Our analyses of genetic mouse models revealed a similar phenotype of defective appendage formation in Gli2/3$^{EKO}$ and Smo$^{EKO}$ mice and uncovered that synergistic signalling by Gli2 and Gli3 is predominantly mediating HH signalling in mammalian skin epithelium (Fig. 6D). Our results also support the previous notion that Gli1 does not seem to play a major role in transducing a HH response in the mammalian epidermis and thus is not a major contributor to appendage formation in mammalian skin.

Our work identifying an essential role for Gli2 in controlling the number and localisation of Lrig1+ve progenitor cells driving SG morphogenesis is of particular interest given that previous data pointed to a role for HH signalling in governing SG differentiation in adult skin. First evidence supporting this hypothesis came from Shh mutant mice that showed deficiency in SG formation (St. Jacques et al, 1998; Chiang et al, 1998). This observation was further supported by a mouse model with epidermis-specific overexpression of an active Smo mutant receptor, resulting in an increase in size and number of SGs in adult skin (Allen et al, 2003). In contrast, blocking HH signalling in epithelial skin cells by

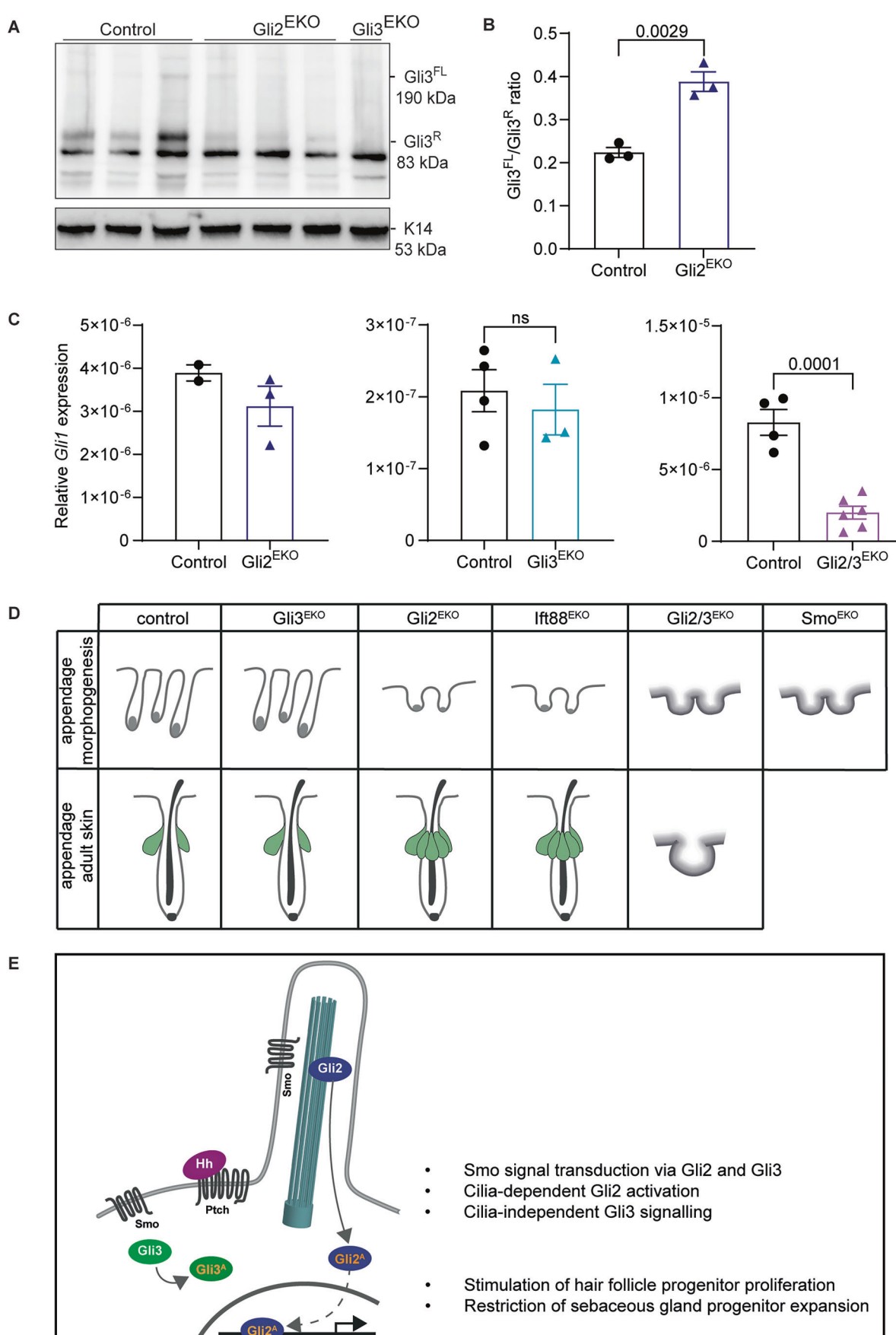

◄ **Figure 6.** **Gli3 activator function in epidermal appendage formation.**

(A, B) Western blot of epidermal samples from P3 Gli2[EKO] and control littermates (A) detecting Gli3 full-length (Gli3[FL]), Gli3 repressor (Gli3[R]) protein and K14 as loading control and quantification of the western blot (B) presenting the Gli3[FL]: Gli3[R] ratios ($n = 3$/genotype). (C) qRT-PCR analysis for Gli1 mRNA expression in tail skin of P6 Gli2[EKO], Gli3[EKO] and Gli2/3[EKO] and control mice. Each datapoint represents one animal ($n = 2$–4 mice/genotype). The results shown are from two technical replicates ($n = 2$ mice/genotype) for Gli1 expression in Gli2/3[EKO] mice. (D) Schematic overview displaying the phenotypic characterisation during skin appendage morphogenesis and in adult skin of different HH signalling mouse mutants, including Gli2[EKO], Gli3[EKO], Ift88[EKO], Gli2/3[EKO] and Smo[EKO] mice. (E) Schematic and mechanistic summary of distinct Gli2 signalling activities in mammalian appendage formation and maintenance. Data were presented as mean ± SEM. *P* value was calculated using an unpaired Student's *t*-test. Source data are available online for this figure.

overexpressing a Gli2 mutant suppresses sebocyte differentiation (Allen et al, 2003). Interestingly, expression of an active Gli2 mutant in the skin epithelium induces a SG duct cell fate and branched and additional SGs in adult mice (Gu and Coulombe, 2008). Furthermore, a role for HH-Gli in differentiation was also detected in a human sebocyte cell line and SG tumours of human and mice (Niemann et al, 2003; Kakanj et al, 2013; Takeda et al, 2006). Together, these studies demonstrate that modulating the level of HH signalling affects SG physiology, how this is mechanistically regulated was not known. Our results suggest that Gli2 could exert differential and context-dependent functions during SG morphogenesis and SG maintenance in adult skin: whereas Gli2 restricts the expansion of SG stem cells during appendage formation, Gli2 and HH activation promote SG differentiation in adult skin and skin tumours (Kakanj et al, 2013). In this context, a recent study identified a role for HH-Gli2 activity for maintaining meibomian glands (MG), specialised SGs in the tarsal plate of the eyelids, and suggested that Gli2 activity can promote progenitor expansion in MG and human MG carcinoma (Zhu et al, 2025). Remarkably, the SG defect observed in Gli2[EKO] mice persists throughout adult life and multiple cycles of hair regeneration, showing that early Gli2-mediated regulation of SG formation is relevant for adult skin physiology.

Our observation that Gli2/3[EKO] mice showed a more severe phenotype when compared to the Gli2 single knockout mouse model strongly suggests a compensatory role for the Gli3 transcription factors in mediating the HH response during appendage formation (Fig. 6D). Moreover, the lowering of the Gli3[R] compared to Gli3[FL] activator in Gli2[EKO] mice points to an under-appreciated Gli3 activator function supporting Gli2 in HF formation and growth.

Given prior reports showing HH responsiveness, particularly in the dermis, our work dissects and highlights HH reception in the skin epithelium. We demonstrate that Gli2 functions, but not Gli3, are cilia-dependent in the process of appendage formation. Intriguing in this respect is that the Ift88 dermal knockout recapitulates the HF growth defect seen in Shh and Gli2 complete knockout mice (Lehman et al, 2009), whereas the Ift88[EKO] mutants phenocopy the milder hair phenotype also observed in Gli2[EKO] animals. Together, these data reveal that either Gli3 does not play a role in the dermal compartment or that cilia play differential roles in transducing the HH signal and are more important in dermal cells. A context-dependent role of primary cilia in transducing the HH signal has been proposed previously, where cilia can act as both positive and negative regulators of the HH signalling pathway. In the neural tube, where Gli activators normally play an important role, defects in cilia and IFT result in loss of function HH phenotypes, whereas in the limb, where Gli3 repressors have a

major role, these defects lead to a gain of function HH phenotype (Haycraft et al, 2005; Huangfu and Anderson, 2005).

Our results suggest that HF and SG development are differentially regulated by the cilia-Gli2 signalling axis and indicate that Gli2 signalling is mediated by cilia. Moreover, given that the skin phenotype is much more severe in Gli2/3[EKO] mice when compared to Ift88[EKO] animals, the results also suggest that Gli3 functions independently from cilia in the mammalian skin epithelium. Our results show that cilia do not play an important role in processing full-length Gli3 into the Gli3[R], further supporting the cilia-independent Gli3 functions in skin appendage formation (Fig. 6E).

## Methods

### Reagents and tools table

| Reagent/resource | Reference or source | Identifier or catalog number |
|---|---|---|
| **Experimental models** | | |
| K14Cre[tg] | Hafner et al, 2004 | |
| Gli2[flox] | Corrales et al, 2006 | |
| Gli3[flox] | Blaess et al, 2008 | |
| Smo[flox] | Long et al, 2001 | |
| Ift88[flox] | Haycraft et al, 2007 | |
| **Antibodies** | | |
| G. Pig anti-K71 | Progen | GP71 |
| G. Pig anti-K75 | Progen | GP-K6HF |
| g. Pig anti-K86 | Progen | GP-HHB6 |
| Rabbit anti-K14 | Biolegend | 905304 |
| Chicken anti-K14 | Biolegend | 906004 |
| Rabbit anti-K10 | Biolegend | 905404 |
| G. Pig anti-K15 | Progen | GPCK15 |
| G. Pig anti-Adipophilin | Fritzgerald Industries | 20R-AP002 |
| Mouse anti-BrdU | BD | 347580 |
| Rat anti-BrdU | OBT | 00306 |
| Goat anti-Lrig1 | R&D | AF3688 |
| Rabbit anti-Filaggrin | Covance | 905801 |
| Rat anti-CD34 | eBioscience | 14-0341-82 |
| Goat anti-Gli3 | R&D | AF3690 |
| Goat anti-Chicken IgY (H + L) Alexa Flour 488 | Invitrogen | A-11039 |

| Reagent/resource | Reference or source | Identifier or catalog number |
|---|---|---|
| Donkey anti-Goat IgG (H + L) Cross-Adsorbed, Alexa Fluor 647 | Invitrogen | A-21447 |
| Goat anti-Rabbit IgG (H + L) Cross-Adsorbed, Alexa Fluor 647 | Invitrogen | A21244 |
| Donkey anti-Rat IgG (H + L) Highly Cross-Adsorbed Alexa Fluor 647 | Invitrogen | A78947 |
| Goat anti-Rat IgG (H + L) Cross-Adsorbed Alexa Fluor 594 | Invitrogen | A-11007 |
| Goat anti-Rabbit IgG (H + L) Cross-Adsorbed Alexa Fluor 594 | Invitrogen | A-11012 |
| Abberior STAR580, Goat anti-Guinea pig IgG | Abberior | ST580-1006 |
| Peroxidase (HRP)- conjugated Donkey anti-Goat IgG (H + L) | Jackson ImmunoResearch | 705-035-003 |
| Peroxidase (HRP)-conjugated Donkey anti-Rabbit IgG | Cytiva | NA934 |
| **Oligonucleotides and other sequence-based reagents** | **Forward primer** | **Reverse primer** |
| qPCR primers | This study | Table 1 |
| Gli2_EX7 | AP7D2KM (TaqMan) | Catalog No. #4331348 |
| **Chemicals, Enzymes and other reagents** | | |
| Formaldehyde | Roth | 7398.1 |
| Hematoxylin & eosin | Roth & Merck | T865.3 & 109844 |
| BrdU | BD, Oxford Biotech | 347580, OBT00306 |
| EDTA | Gibco | 15575 |
| PBS | AppliChem | A0964 |
| Goat serum | Sigma-Aldrich | G9023 |
| Donkey serum | Sigma-Aldrich | D9663 |
| Milk powder | Roth | T145.1 |
| Fish skin gelatin | Sigma-Aldrich | 935425 |
| Triton X-100 | Sigma-Aldrich | T8787 |
| DAPI | Sigma-Aldrich | D9542 |
| Nile red | Sigma-Aldrich | N3013 |
| Collagenase | Gibco | 17018029 |
| DNase | Millipore | D5025 |
| Trypsin | Life Technologies GmbH | 15090046 |
| F12 | Gibco | 12765 |
| DMEM | Sigma-Aldrich | D5671 |
| FBS Superior | Merck Millipore | S0615 |
| BSA | Sigma-Aldrich | A2153 |
| Ammonium thiocyanate | Sigma-Aldrich | A7149 |
| ß-Mercaptoethanol | Sigma-Aldrich | 63689 |
| 4–15% gradient gels | Bio-Rad | 4561084 |
| PVDF membrane | GE Healthcare | 88518 |
| Ponceau S | Sigma-Aldrich | P3504 |

| Reagent/resource | Reference or source | Identifier or catalog number |
|---|---|---|
| 2x Laemmli | Bio-Rad | 1610737 |
| **Software** | | |
| Adobe Illustrator CS5 | Adobe | |
| Cell Ranger v9.0.1 | 10x Genomics | |
| Fiji | Schindelin et al, 2012 | |
| Prism | GraphPad | |
| Seurat v5.1.0 | https://doi.org/10.1038/s41587-023-01767-y | |
| RunHarmony v1.2.1 | https://doi.org/10.1101/461954 | |
| Metascape | metascape.org | |
| **Other** | | |
| BX53 microscope | Evident Scientific | |
| IX83 | Evident Scientific | |
| FV1000 | Evident Scientific | |
| Stellaris5 | Leica Microsystems | |
| 7900HT Sequence Detection System | Applied Biosystems | |
| Illumina NovaSeq 6000 | Illumina | |
| Quantstudio3 | Applied Biosystems | |
| Chromium Next GEM Single Cell 3' Kit v4 | 10x Genomics | PN - 1000747 |
| KAPA Library Quantification Kit Complete kit (ABI Prism) | Roche | KK4835 |
| RNAMagic | My-Biobudget | 56-1000-100 |
| RNeasy Fibrous Tissue Mini Kit | Qiagen | 74704 |
| Quantitech reverse transcription kit | Qiagen | 205313 |
| Amersham Prime ECL | Cytiva | |
| Chemiluminescent substrate for Western Blotting | Cyanagen | PH01A-LB-2 |
| Cell strainer | Avantor | 732-2758/59 |

## Experimental mice

Epidermal deletion of Gli2, Gli3, Smo and Ift88 was accomplished by crossing K14Cre[tg] mouse line (Hafner et al, 2004) with Gli2[flox] (Corrales et al, 2006), Gli3[flox] (Blaess et al, 2008), Smo[flox] (Long et al, 2001) and Ift88[flox] (Haycraft et al, 2007) lines which have been described previously. All mouse strains except Smo[flox] and Ift88[EKO] were maintained and crossed on the C57BI/6N background. Ift88[EKO] mice were maintained and crossed on a mixed FVB/N and C57Bl/6N background. Floxed littermates without Cre recombinase were used as a control without gender preference. For labelling proliferating cells, a single dosage of BrdU (Sigma; 100 mg/kg body weight) was administered i.p. 1 h prior to sacrificing the animals. All husbandry and animal experiments were conducted according to the guidelines and license approval by the State Office of North Rhine-Westphalia, Germany.

**Table 1. List of mouse primers for quantitative real-time PCR (qRT-PCR).**

| Target | Forward primer | Reverse primer |
| --- | --- | --- |
| 18S | 5′- ATCAGATACCGTCGTAG -3′ | 5′- GCAAAGCTGAAACTTAAAG-3′ |
| Filaggrin | 5′- GACAGCCAAGTCCATTCT-3′ | 5′- ACTCATTCCTCCCTGAC-3′ |
| Gli1 | 5′-GCACCACATCAACAGTGAGC-3′ | 5′-GACTTCCGACAGCCTTCAAA-3′ |
| Gli2 | 5′- CGCATGATTCGGACCTCTC-3′ | 5′- GATTGATGGGGTGGGGAAAA-3′ |
| Gli3 | 5′- AACTCCTTGGTTACAATCCTCAAT-3′ | 5′- GATAGGTCTCTGTGTTGGAAATGT-3′ |
| K10 | 5′- ACCCTTAGCAAGTCTGACCT-3′ | 5′- CGTTCATTTCCACATTCACATCAC-3′ |
| K71 | 5′- AGACTCCGCTCAGAGATTG-3′ | 5′- CATCCTTGAGGGCACTGT-3′ |
| K75 | 5′- CGCAACACCAAACAAGAG-3′ | 5′- GGCATCCTTGAGAGCTAG-3′ |
| K86 | 5′- GGAGCAGAGGTTGTGTGAGG-3′ | 5′- AGGGGCAGTACCAGAGACG-3′ |
| K15 | 5′- TGGAGATGCAGATTGAGCAGCTGAA-3′ | 5′- TGCTCCCTCATCTCTGCCAGCA-3′ |
| CD34 | 5′- GGACAGCAGTAAGACCACACCAGC-3′ | 5′- CCCCAACTGGCATACTGCCT-3′ |

## Histological analyses

Harvested back skin samples were fixed in 4% formaldehyde solution for 2 h. Paraffin-embedded samples were sectioned with a 5-µm thickness, and hematoxylin & eosin (H&E) staining was performed. Images were taken with a BX53 microscope (Evident Scientific), analysed with Fiji software and scale bars were added using the Fiji software.

For quantification of HF number, HFs with attached SG or dermal papillae were counted from three different fields on each skin sample. HF numbers were normalised to the covered skin area.

For hair type analysis, hairs from three different areas along the central axis were plugged and mounted on slides. A total of 100 hairs per animal were counted. Hair types were determined based on the patterns and bends of the hair shaft. For each genotype, at least three mice were analysed.

Quantification of HF at distinct stages of HF morphogenesis (stages 4, 5 and 6) was done on back skin sections from P0 Gli2$^{EKO}$, Gli3$^{EKO}$, Gli2/3$^{EKO}$ and control mice, and percentages were calculated relative to total HF number. According to Saxena et al, 2018, the following criteria applied for stage 4: the pre-mature hair placode is invaginating into dermis with the dermal condensate closely associated with the growing tip of the placode. At stage 6, when the HF looks more mature, the hair matrix is beginning to surround the dermal condensate, and a hair shaft starts to appear.

## Isolation of the tail epidermal whole mount

As described previously (Braun et al, 2003), tail skin was incubated with 5 mM EDTA (Gibco) for 1 h (P0 animals), 1 h 20 min (P3 animals), 1 h 30 min to 2 h (P6 animals) and 3 h (for older animals) at 37 °C. The tail epidermis was removed carefully with a tweezer and fixed in 3.4% formaldehyde solution for 1 h, followed by two times PBS wash. Tail epidermis samples were stored in PBS at 4 °C for further analysis.

## Immunofluorescence staining

For immunofluorescence, paraffin tissue sections were incubated with 10% normal goat/donkey serum or 1% milk powder for 1 h. For K71, K75 and K86 staining, 0.1% Triton X was included in the blocking solution. Epidermal whole mounts were incubated with TB Buffer (0.25% Fish Skin Gelatin, 0.005 Triton X, 0.25% milk powder in TBS) for 1 h. The following primary antibodies were used: K14 (rabbit, 1:1000; Covance), K10 (rabbit, 1:1000; Covance), K71 (g.pig, 1:200, Progen), K75 (g.pig, 1:200, Progen), K86 (g.pig, 1:200, Progen), K15 (g.pig, 1:100, Progen), Adipophilin (g.pig, 1:250, Fitzgerald Industries), BrdU (mouse, 1:20, BD; rat, 1:500; Oxford Biotech), Lrig1 (goat, 1:100, R&D), Filaggrin (rabbit, 1:1000, Covance) and CD34 (rat, 1:50, eBioscience). All secondary antibodies used were Alexa-488, Alexa-647, Alexa-594 (1:500, Invitrogen) and Abberior STAR580 (1:500, Abberior). 1:1000 DAPI (20 mg/ml, Sigma-Aldrich) was used to counterstain the nuclei.

Nile red solution (1 mg/ml in acetone) was freshly diluted 1:1000 in PBS, and tail whole mounts were incubated for 30 min at RT before rinsing a few times with PBS. For nuclei counterstaining, DAPI was used. All images were taken on widefield (IX83, Evident Scientific), confocal (FV1000, Evident Scientific and Stellaris5, Leica Microsystems) and brightfield (BX53, Evident Scientific) microscopes and analysed with Fiji software. Background subtraction and changes to brightness and contrast were done according to the image signal, and scale bars were added using the Fiji software.

## Quantification of immunofluorescence stainings

### Quantification of BrdU$^+$ cells
BrdU$^+$ cells in back skin sections from P0 control, Gli2$^{EKO}$, Gli3$^{EKO}$ and Gli2/3$^{EKO}$ were counted together with the total number of cells (DAPI$^+$) and their ratio was calculated as the percentage of BrdU$^+$ cells. For quantification of the number of BrdU$^+$ cells in the Lrig1-compartment, BrdU$^+$/Lrig1+ double-positive cells were counted manually in tail whole mounts from P0 control and Gli2$^{EKO}$ mice. Only the central HFs in HF triplets were considered, and 12–22 HFs per animal were counted.

### Quantification of sebaceous gland size and number
Following Nile red/Adipophilin staining, tail whole mounts from P49 and P6 control, Gli2$^{EKO}$ and Ift88$^{EKO}$ mice were imaged at least at three different fields using FV1000 (Evident Scientific). The individual size of the sebaceous lobes was measured with Fiji software, focusing on the Nile red/Adipophilin positive area. SGs

with low integrity and no clear separation were excluded from the analysis. Based on the SG staining, the number of individual glands per HF was calculated.

### Quantification of K10 and filaggrin

Fluorescence images of the back skin sections form P6 and P49 control and Gli2/3$^{EKO}$ mice were analysed using custom Fiji macros. The intensity was quantified using a macro to process files containing predefined regions of interest (ROIs), including background subtraction. The ROIs corresponded to the area between the HF infundibulum and the distal tip of the HF within the skin tissue. To exclude potential artifacts, a mask was generated by thresholding the images to remove pixels with intensity values exceeding 8000 arbitrary units (AU) for K10 and 11000 AU for filaggrin. The filtered images were then quantified within the ROIs to extract mean fluorescence intensity.

### Quantification of K15 and CD34

Fluorescence images of the back skin sections form P6 and P49 control and Gli2/3$^{EKO}$ mice were examined for the presence of K15 or CD34 in HF structures, and HF were classified as either positive or negative. Only regions of HFs where K15 or CD34 expression is typically expressed were analysed.

### Quantification of Lrig1+ cells

Lrig1$^{+}$ cells in HFs of tail whole mounts from P6 control and Gli2$^{EKO}$ mice were counted manually. About 7–10 central HFs in HF triplets were counted per animal

### Quantification of the length and thickness of tail skin HFs

Quantification of HFs in tail whole mounts from P3 control and Gli2$^{EKO}$ was done by setting ROIs corresponding to the length and thickness of HFs. The ROIs were measured using a macro on the maximum projection of the z-stack images. Both macros operated in a fully automated manner, and all measurement outputs were exported as CSV files for downstream analysis. About 7–16 central HFs in HF triplets per animal were analysed.

## Sample preparation for single-cell RNA sequencing

Epidermal whole mounts were isolated from the back skin of P2 control and Gli2$^{EKO}$ mice. Therefore, skin was cut into small pieces of 1 cm$^2$ and incubated dermis-side down in Collagenase (2 mg/ml, Gibco)/DNase (1 U/ml, Millipore) for 1 h 15 min at 37 °C. Residues of digested dermis were removed by gently pipetting PBS over the skin tissues. The epidermal whole mounts were then incubated in 0.25% Trypsin (Life Technologies GmbH)/0.05 mM EDTA (Gibco) for 1 h at 37 °C, with gentle shearing every 15 min. Trypsin was neutralised with 500 μl of FAD media (F12, Gibco + DMEM, Sigma-Aldrich + 10% FBS Superior, Merck Millipore). The cell suspension was filtered through a 30 μm cell strainer (Avantor), which was subsequently washed with an additional 500 μl of media to recover remaining cells. Cells were centrifuged and resuspended in 0.004% BSA/PBS. Samples with a viability of more than 75% were processed for sequencing.

## Single-cell RNA sequencing

Single cell suspensions in 0.004% BSA/PBS (700-1200 nuclei/μL) were processed with the Chromium Next GEM Single Cell 3′ Kit v4 (10X Genomics) with dual indices, aiming for a target of 5000 cells/sample while using the on-chip multiplexing option. Cells and the appropriate mastermix were loaded on a Chromium Next GEM Chip G and run on the Chromium X (10X Genomics) to generate Gel Bead-In-EMulsions (GEMs) according to the manufacturer's protocol. After incubation, the GEMs were broken, and the pooled fractions were recovered. Silane magnetic beads were used to remove leftover biochemical reagents and primers from the post-GEM reaction mixture. Full-length, barcoded cDNA was then amplified by PCR to generate sufficient amounts for library construction. Library preparation was performed, including End Repair, A-tailing, Adaptor Ligation and PCR. The final libraries were quantified (Qubit) and validated (Tape Station), pooled, and the library pool was then quantified using the Peqlab KAPA Library Quantification Kit (Roche) and the Applied Biosystems 7900HT Sequence Detection System. Libraries were sequenced on an Illumina NovaSeq 6000 sequencing instrument with a 29 + 89 bp sequencing protocol.

## Single-cell RNA sequencing data analysis

Fastq files from a 10x Single Cell 3′ v4 (polyA) OCM run were processed using the Cell Ranger v9.0.1 software against the GRCm39 reference and GENCODE Mouse Release M30 gene annotation. The Cell Ranger default filtered_feature_bc_matrix folder for each genotype sample were further filtered in Seurat v5.1.0 (https://doi.org/10.1038/s41587-023-01767-y) to include cells with more than 1000 unique features but less than 8000 and a mitochondrial percentage below 15. The Bioconductor.org package, scDblFinder v1.18.0 (https://doi.org/10.12688/f1000research.73600.2) was used to identify doublets, which were also removed. The resulting matrix used for downstream analysis consisted of 7199 cells (3009 Gli2$^{EKO}$, 4190 control) with a median of 3809 features per cell.

All data was Log-Normalised and the top 2000 variable genes scaled, and their dimensionality reduced using PCA. Data from both genotypes were integrated using the default settings of the RunHarmony v1.2.1 function (https://doi.org/10.1101/461954). UMAP was used for downstream dimensionality reduction from the top 20 principal components (PCs). The top 20 PCs were also used for the construction of the shared nearest neighbours graph using Seurat's FindNeighbors function and subsequent clustering using FindClusters function with the Louvain algorithm with multilevel refinement and a resolution parameter of 0.5.

To focus on Keratinocytes, we first identified all clusters using known canonical markers. The following clusters of cells were labelled for removal: clusters expressing Fbn1 were labelled as fibroblasts, Rgs5 as Smooth Muscles, Pmel as melanocytes, Pecam1 as endothelial cells, and lastly, Fcer1g, Lcp1 as immune cells. This left 5291 keratinocytes (2190 Gli2$^{EKO}$ and 3101 Control), which were processed in the same way as the full dataset.

All differential expression was performed using the Wilcoxon rank-sum test in the Seurat FindMarkers function. The criteria for the tested genes were that they must be expressed in 25% of the respective population and have a log-foldchange $\geq$ 0.25. $P$ value adjustments for multiple comparisons was performed using the Bonferroni correction.

## Gene ontology analysis

Gene ontology (GO) analysis of clusters 8 and 13 was performed using the online Gene Annotation and Analysis Resource tool

Metascape (metascape.org). The list of top 110–130 down- and upregulated genes per cluster was selected, and GO was performed for pathway and process enrichment analysis. Relevant pathways and processes were presented along the fold enrichment, $-\log10(P)$.

## qRT-PCR analysis

RNA isolation was done using RNAMagic (Biobudget) and RNeasy Fibrous Tissue Mini Kit (Qiagen). cDNA was synthesised from 0.5 µg total RNA using the Quantitech reverse transcription kit (Qiagen) according to the manufacturer's protocol. All primers used for qRT-PCR are listed in Table 1. Three technical replicates per biological sample were run on Quantstudio3 (Applied Biosystems). Relative gene expression was normalised to 18S housekeeping gene expression.

## Protein analysis by Western blot

Back skin of P3 Control, Gli2$^{EKO}$ and Gli3$^{EKO}$ was used to separate epidermis and dermis. Skin tissue was incubated for 30 min in 400 µl ice-cold split buffer containing 3.8% ammonium thiocyanate. To prepare lysates for WB, epidermal sheets were washed in PBS and added to 50 µl 2x Laemmli (Bio-Rad) buffer before dissociating the tissue by pipetting and freezing on dry ice. Lysates were supplemented with 10% ß-mercaptoethanol and incubated for 5 min at 95 °C before loading a 10 µl sample on 4–15% gradient gels (Bio-Rad). SDS page was performed for 1 h at 100 V. Proteins were transferred onto PVDF membranes (0.45 µm pore size, GE Healthcare) for 16 h at 35 V. Membranes were stained for 5 min with Ponceau S (Sigma) before blocking with 5% milk in TBST (Tris-buffered saline with Tween 20) for 1 h at RT. Membranes were probed with primary anti-Gli-3 antibody (1:1000, R&D) overnight at 4 °C, followed by washing steps and incubation with the secondary HRP-conjugated anti-goat antibody for 45 min at RT (1:10,000, Jackson ImmunoResearch). For detection of K14, membranes were incubated with anti-K14 (1:10,000, rabbit, BioLegend) overnight at 4 °C, before incubating with anti-rabbit pig HRP (1:10,000, Cytiva) antibody for 45 min at RT. Detection of signal was performed with Amersham Prime ECL (Cytiva). Quantification of Gli3$^{FL}$, Gli3$^R$ and K14 band intensities was performed with Fiji, and the ratio between proteins was calculated.

## Statistical analysis

Statistical significance was calculated using an unpaired two-tailed Student's *t*-test, except for data presented in Fig. EV2B, where a paired two-tailed *t*-test was performed. All data were described as mean ± SEM or ±SD as indicated. All statistical analyses were done using GraphPad Prism 10 (GraphPad Software, Inc., La Jolla, CA, USA). Experiments were analysed in a non-blinded manner.

## Data availability

The data discussed in this publication have been deposited in NCBI's Gene Expression Omnibus and are accessible through GEO Series accession number GSE 294647 (https://www.ncbi.nlm.nih.gov/geo/query/acc.cgi?acc=GSE294647).

The source data of this paper are collected in the following database record: biostudies:S-SCDT-10_1038-S44318-025-00519-9.

## Peer review information

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

## Acknowledgements

We are grateful to Sandra Blaess (University of Bonn) for providing floxed Gli mouse lines and Beate Lichtenberger (University of Vienna) for the floxed smoothened mouse line. We thank Houda Khatif for excellent experimental support and Deborah Delbue for assisting in manuscript preparation. We thank for the support by the animal facilities of CECAD, CMMC and the Medical Faculty of the University of Cologne. This work was supported by the DFG

Research Infrastructure West German Genome Center (project 407493903) as part of the Next Generation Sequencing Competence Network (project 423957469). NGS analyses were carried out at the production site Cologne (Cologne Center for Genomics (CCG)). We like to acknowledge the Center for Molecular Medicine Cologne (CMMC) Microscopy Service, with special thanks to Mehrnaz Babaki for supporting imaging and analysis. The Leica STELLARIS5 confocal microscope was funded by the German Research Foundation (DFG-INST 216/1174-1 FUGG). The work of H. Bazzi was funded by the German Research Foundation (SFB829, Project-ID 73111208). Research of C. Niemann was funded by the LEO Foundation (LF-OC-22-001112) and the German Research Foundation (SFB829, Project-ID 73111208 and DFG grant, Project-ID 507956072). C Niemann is grateful for financial support from the Köln Fortune Programme of the Medical Faculty (Project-ID 342/2019) and the CMMC of the University of Cologne.

## Author contributions

**Gokcen Gozum**: Conceptualisation; Formal analysis; Validation; Investigation; Methodology. **Lakshit Sharma**: Formal analysis; Validation; Investigation; Visualisation; Methodology. **Paula Henke**: Formal analysis; Validation; Investigation; Visualisation; Methodology. **Lisa Wirtz**: Formal analysis; Validation; Investigation. **Mareike Damen**: Formal analysis; Investigation. **Viktoria Reckert**: Formal analysis; Investigation. **Peter Schettina**: Investigation. **Melanie Nelles**: Investigation. **Craig N Johnson**: Data curation; Software; Formal analysis; Visualisation. **Hisham Bazzi**: Conceptualisation; Resources; Supervision; Writing—original draft; Writing—review and editing. **Catherin Niemann**: Conceptualisation; Resources; Supervision; Funding acquisition; Visualisation; Writing—original draft; Project administration; Writing—review and editing.

Source data underlying the figure panels in this paper may have individual authorship assigned. Where available, figure panel/source data authorship is listed in the following database record: biostudies:S-SCDT-10_1038-S44318-025-00519-9.

## Funding

## Disclosure and competing interests statement

The authors declare no competing interests.

# Expanded View Figures

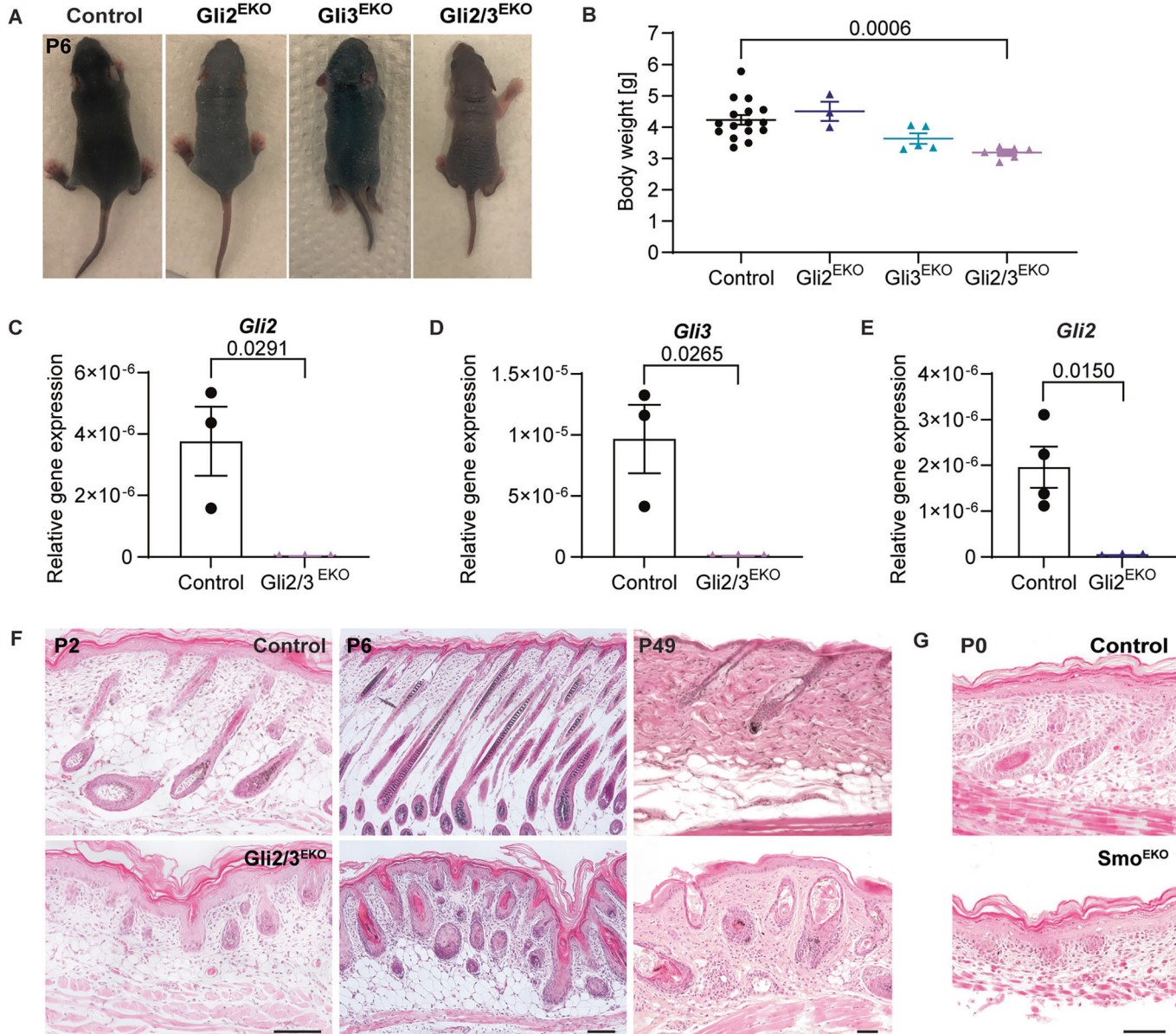

**Figure EV1. Characterisation of Gli^EKO and Smo^EKO mouse models.**

(A, B) Representative images (A) and body weight (B) of P6 Gli2^EKO, Gli3^EKO, Gli2/3^EKO and control mice. ($n = 3$-16mice/genotype). (C, D) qRT-PCR analysis for Gli2 (C) and Gli3 (D) mRNA expression in tail epidermis from P6 Gli2/3^EKO and control littermates. Each datapoint represents one animal. ($n = 3$ mice/genotype). (E) qRT-PCR analysis for Gli2 mRNA expression in back skin epidermis from P3 Gli2^EKO and control littermates. Each datapoint represents one animal. ($n = 3$-4 mice/genotype). (F) Representative H&E staining of back skin sections from P2, P6 and P49 Gli2/3^EKO and control littermates ($n = 3$ mice/genotype). (G) Representative H&E staining of back skin sections from P0 Smo^EKO and control littermates. Each datapoint represents one animal. ($n = 3$ mice/genotype). Scale bars 50 μm (F, G). Data were presented as mean ± SEM. *P* value was calculated using an unpaired Student's *t*-test.

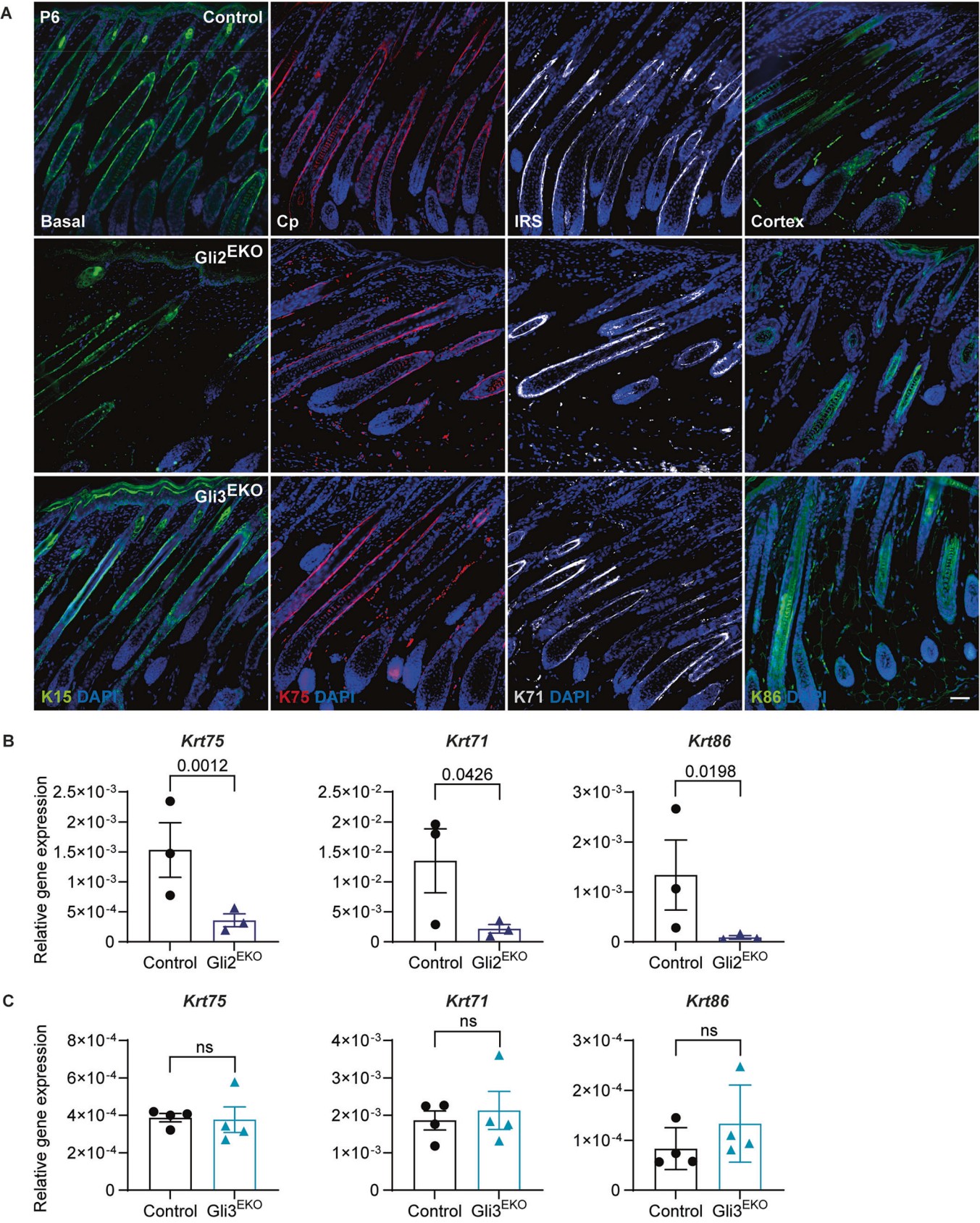

◀ **Figure EV2. Analysis of hair lineage differentiation in Gli2^EKO and Gli3^EKO mice.**

(A) Immunofluorescence staining for K15 (green), K75 (red), K71 (grey), K86 (green) and DAPI (blue, nuclei) of back skin sections from P6 Gli2^EKO, Gli3^EKO and control mice. ($n = 3$ mice/genotype) (redisplay of data for control mice from Fig. 2A). (B, C) qRT-PCR analysis of Krt75, Krt71 and Krt86 mRNA expression in back skin of P6 Gli2^EKO (B) and Gli3^EKO (C) and control littermates. ($n = 3–4$ mice/genotype). Scale bar 50 μm. Data were represented as mean ± SEM. *P* value was calculated using paired (B) and unpaired Student's *t*-test (C).

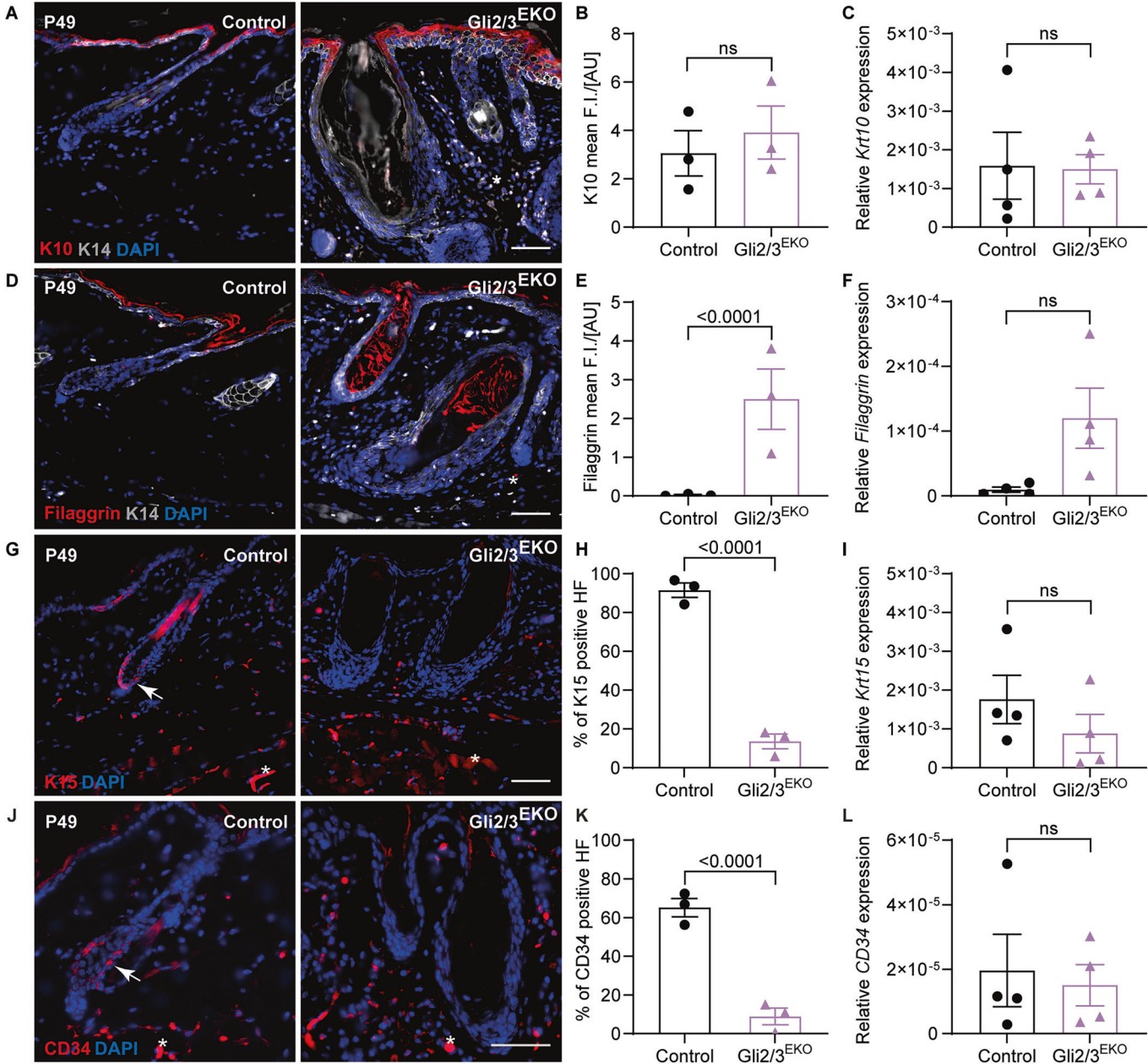

**Figure EV3.    Defective hair differentiation in Gli2/3^EKO mice.**

(A, B) Immunofluorescence staining for K10 (red), K14(grey) and DAPI (blue, nuclei) (A) and quantification of K10 (B) in back skin sections from P49 Gli2/3^EKO and control littermates ($n = 3$ mice/genotype). K10 fluorescence intensity (FI) per arbitrary unit (AU). (C) qRT-PCR analysis of Krt10 mRNA expression in back skin from P49 Gli2/3^EKO and control littermates. The results shown are from two technical replicates ($n = 2$ mice/genotype). (D, E) Immunofluorescence staining for Filaggrin (red), K14 (grey) and DAPI (blue, nuclei) (D) and quantification of filaggrin (E) in back skin sections from P49 Gli2/3^EKO and control littermates ($n = 3$ mice/genotype). Filaggrin fluorescence intensity (FI) per arbitrary unit (AU) ($p$ value = 8.42e-06). (F) qRT-PCR analysis of filaggrin mRNA expression in back skin from P49 Gli2/3^EKO and control littermates. The results shown are from two technical replicates ($n = 2$ mice/genotype). (G, H) Immunofluorescence staining for K15 (red, arrow) and DAPI (blue, nuclei) (G) and quantification of K15 (H) in back skin sections from P49 Gli2/3^EKO and control littermates. 50–147 HFs per animal were counted. ($n = 3$ mice/genotype) ($p$ value = 5.4e-17). (I) qRT-PCR analysis of K15 mRNA expression in back skin from P49 Gli2/3^EKO and control littermates. The results shown are from two technical replicates ($n = 2$ mice/genotype). (J, K) Immunofluorescence staining for CD34 (red, arrow) and DAPI (blue, nuclei) (J) and quantification of CD34 (K) in back skin sections from P49 Gli2/3^EKO and control littermates. 106–175 HFs per animal were counted. ($n = 3$ mice/genotype) ($p$ value = 8.17e-24). (L) qRT-PCR analysis of CD34 mRNA expression in back skin form P49 Gli2/3^EKO and control littermates. The results shown are from two technical replicates ($n = 2$ mice/genotype). Scale bars 50 μm (A, D, G, J). An asterisk marks an unspecified signal (G, J). Data were represented as mean ± SEM. $P$ value was calculated using an unpaired Student's $t$-test.

A

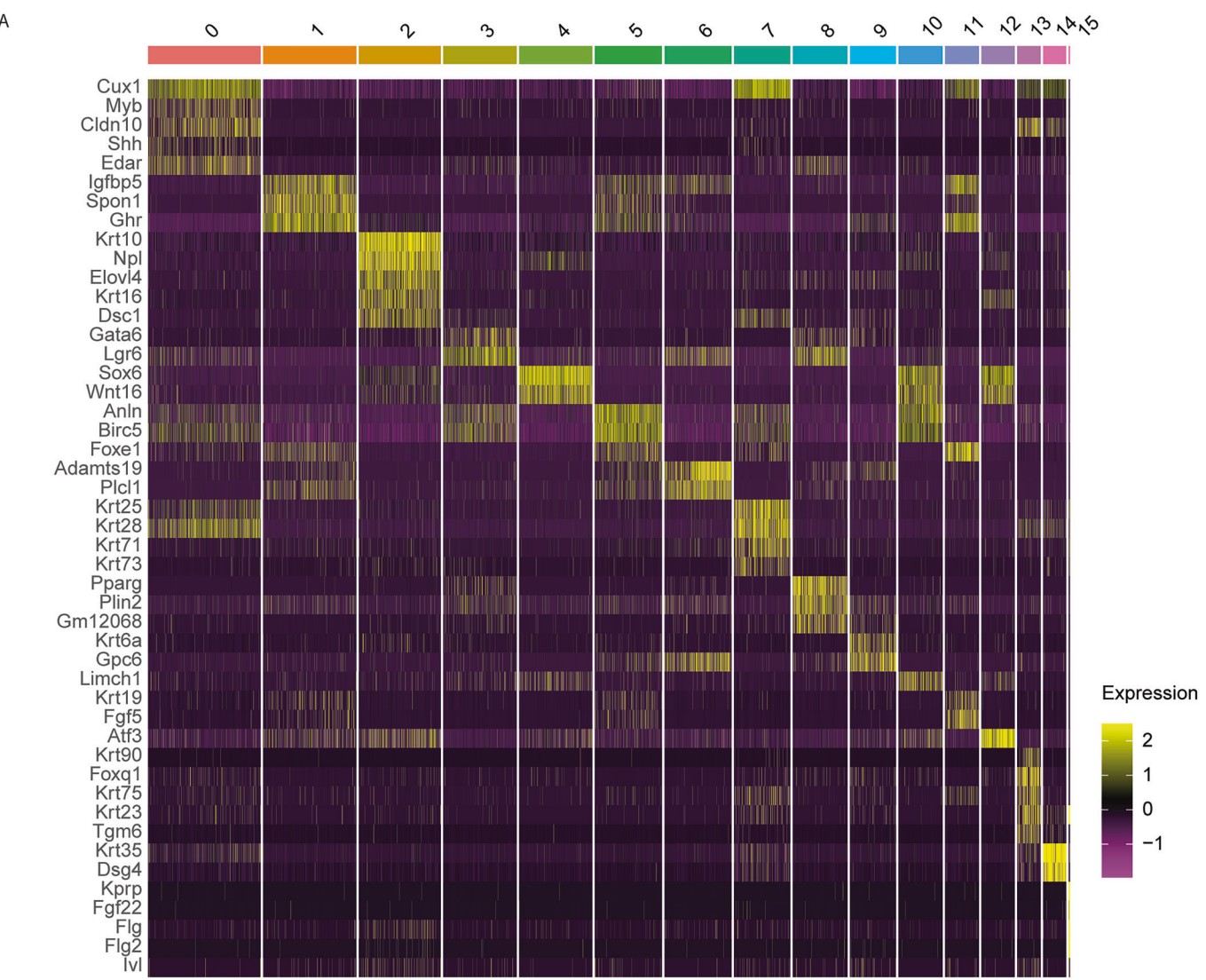

**Figure EV4. Marker expression in keratinocyte clusters.**

(A) Heatmap showing the marker expression defining each cluster. Colours represent scaled expression by gene.

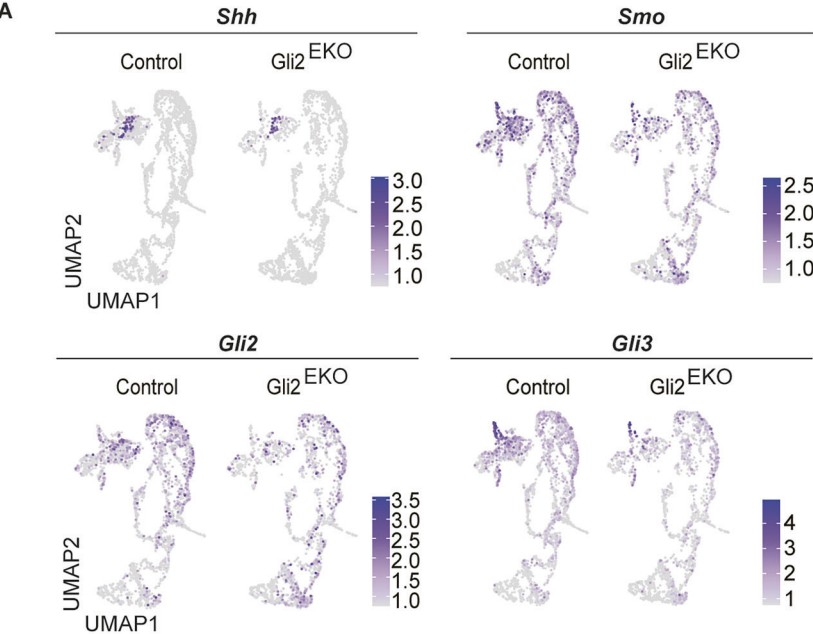

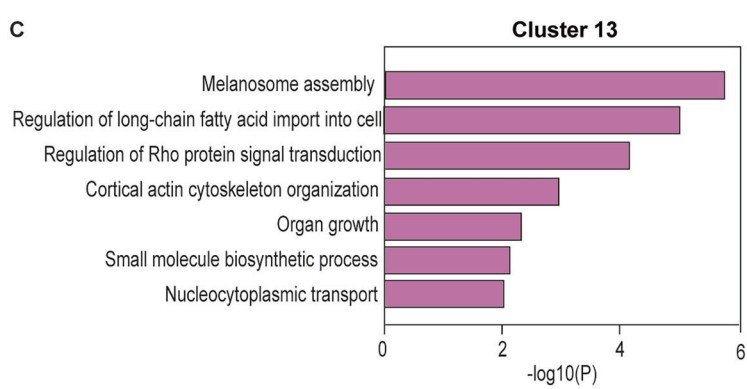

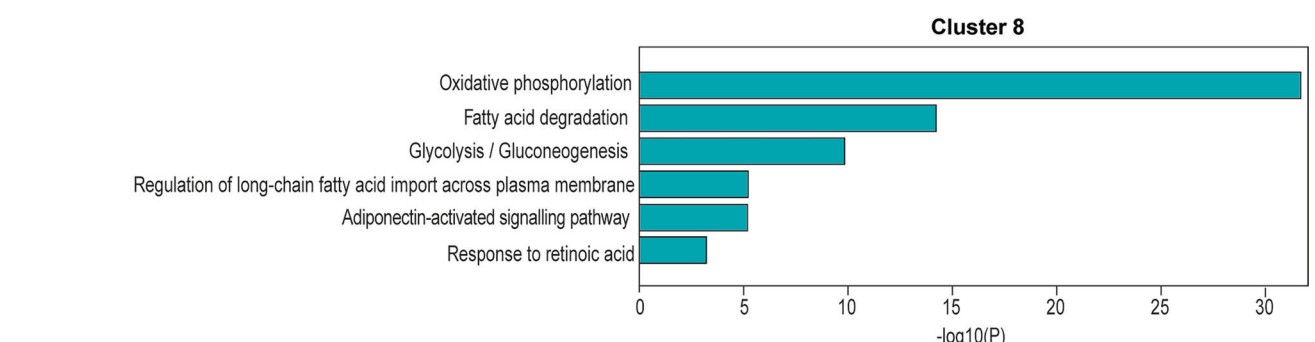

**Figure EV5. Molecular and cellular characterisation of distinct keratinocyte populations in Gli2EKO mice.**

(A) Feature plots of scRNAseq data showing expression of Shh, Smo, Gli2, and Gli3 in Control vs Gli2EKO keratinocytes. Expression levels are colour coded, and expression is shown for values greater than 0.5. (B) Numbers and percentages of keratinocytes for clusters 0, 7, 13, 14 and 8 in control vs. Gli2EKO scRNAseq dataset. (C, D) Gene ontology for pathway and process enrichment analysis of the top 110–130 down- and up-regulated genes in clusters 13 (C) and 8 (D), respectively. P values were calculated based on the cumulative hypergeometric distribution.

