## [Peer Review File · The EMBO Journal]

Specific and redundant roles for Gli2 and Gli3 in establishing cell fate during murine hair follicle development

Catherin Niemann, Gokcen Gozum, Lakshit Sharma, Paula Henke, Lisa Wirtz, Mareike Damen, Viktoria Reckert, Peter Schettina, Melanie Nelles, Craig Johnson, and Hisham Bazzi

Corresponding author: Catherin Niemann (cnieman1@uni-koeln.de)

Review Timeline:

Submission Date:	5th Oct 24
Editorial Decision:	4th Nov 24
Revision Received:	18th Apr 25
Editorial Decision:	16th May 25
Revision Received:	30th May 25
Accepted:	4th Jul 25

Editor: Daniel Klimmeck

Transaction Report:

Dear Dr Niemann,

Thank you for the submission of your manuscript (EMBOJ-2024-119233) to The EMBO Journal. Please accept my apologies for getting back to you with delay due to protracted referee input and detailed discussion in the editorial team. As mentioned earlier, your study was assessed by two reviewers with expertise in skin biology and hair follicle signaling, whose comments are enclosed below.

As you will see from the experts' reports, the referees acknowledge the analysis and potential interest of your results. However, they also express major concerns regarding completeness and robustness of the findings, which need to be addressed thoroughly to make them supportive of publication in the EMBO Journal. The reviewers also raise issues related to the data presentation, additional controls and improved methods annotation required, statistics applied and overall discussion of related literature, that would need to be conclusively addressed to achieve the level of robustness and clarity needed for The EMBO Journal.

Given the overall interest stated and broader angle of your findings, we are able to invite you to revise your manuscript experimentally to address the referees' comments. I need to stress though that we do require strong support from the referees on a revised version of the study in order to move on to publication of the work.

Please feel free to contact me if you have any questions or need further input on the referee comments.

When submitting your revised manuscript, please carefully review the instructions below.

Please feel free to approach me any time should you have additional questions related to this.

Thank you for the opportunity to consider your work for publication.

I look forward to your revision.

Best regards,

Daniel Klimmeck

Daniel Klimmeck, PhD
Senior Editor
The EMBO Journal

Instruction for the preparation of your revised manuscript:

- 1) a .docx formatted version of the manuscript text (including legends for main figures, EV figures and tables). Please make sure that the changes are highlighted to be clearly visible.
- 2) individual production quality figure files as .eps, .tif, .jpg (one file per figure).
- 3) a .docx formatted letter INCLUDING the reviewers' reports and your detailed point-by-point response to their comments. As part of the EMBO Press transparent editorial process, the point-by-point response is part of the Review Process File (RPF), which will be published alongside your paper.
- 4) a complete author checklist, which you can download from our author guidelines ([https://wol-prod-cdn.literatumonline.com/pb-assets/embo-site/Author Checklist%20-%20EMBO%20J-1561436015657.xlsx](https://wol-prod-cdn.literatumonline.com/pb-assets/embo-site/Author%20Checklist%20-%20EMBO%20J-1561436015657.xlsx)). Please insert information in the checklist that is also reflected in the manuscript. The completed author checklist will also be part of the RPF.
- 5) Please note that all corresponding authors are required to supply an ORCID ID for their name upon submission of a revised

manuscript.

6) It is mandatory to include a 'Data Availability' section after the Materials and Methods. Before submitting your revision, primary datasets produced in this study need to be deposited in an appropriate public database, and the accession numbers and database listed under 'Data Availability'. Please remember to provide a reviewer password if the datasets are not yet public (see <https://www.embopress.org/page/journal/14602075/authorguide#datadeposition>).

7) Our journal encourages inclusion of *data citations in the reference list* to directly cite datasets that were re-used and obtained from public databases. Data citations in the article text are distinct from normal bibliographical citations and should directly link to the database records from which the data can be accessed. In the main text, data citations are formatted as follows: "Data ref: Smith et al, 2001" or "Data ref: NCBI Sequence Read Archive PRJNA342805, 2017". In the Reference list, data citations must be labeled with "[DATASET]". A data reference must provide the database name, accession number/identifiers and a resolvable link to the landing page from which the data can be accessed at the end of the reference. Further instructions are available at .

8) At EMBO Press we ask authors to provide source data for the main and EV figures. Our source data coordinator will contact you to discuss which figure panels we would need source data for and will also provide you with helpful tips on how to upload and organize the files.

Numerical data can be provided as individual .xls or .csv files (including a tab describing the data). For 'blots' or microscopy, uncropped images should be submitted (using a zip archive or a single pdf per main figure if multiple images need to be supplied for one panel). Additional information on source data and instruction on how to label the files are available at .

9) We replaced Supplementary Information with Expanded View (EV) Figures and Tables that are collapsible/expandable online (see examples in <https://www.embopress.org/doi/10.15252/embj.201695874>). A maximum of 5 EV Figures can be typeset. EV Figures should be cited as 'Figure EV1, Figure EV2' etc. in the text and their respective legends should be included in the main text after the legends of regular figures.

11) For data quantification: please specify the name of the statistical test used to generate error bars and P values, the number (n) of independent experiments (specify technical or biological replicates) underlying each data point and the test used to calculate p-values in each figure legend. The figure legends should contain a basic description of n, P and the test applied. Graphs must include a description of the bars and the error bars (s.d., s.e.m.).

We realize that it is difficult to revise to a specific deadline. In the interest of protecting the conceptual advance provided by the work, we recommend a revision within 3 months (2nd Feb 2025). Please discuss the revision progress ahead of this time with the editor if you require more time to complete the revisions.

Referee #1:

The manuscript by Gozum et al represents a mouse genetics tour de force where the authors employ several single and double knockout mice to test the role and mechanisms of action of hedgehog (HH) signaling in skin epithelium, specifically in the formation of hair follicle and sebaceous gland. The authors find that Gli3 has overlapping roles with Gli2 that correspond with their function in the cilium. Gli3 also has a cilium independent function, as Gli12/3 dKO mice phenotype is not recapitulated in mutant mice lacking the cilium in the skin epithelium. When HH signaling is fully perturbed (such is the Gli2/3 dKO mice or Smo KO mice) the hair follicle morphogenesis is impaired, with multiple defects including: impaired hair follicle stem cell formation, impaired proliferation, lack of hair follicle differentiated lineages, adoption of what appears an epidermis like phenotype in the hair, with cyst-like structures taking the place of the hair follicle. This lack of hair lineages in the Gli2/3 mutants seems to be compensated by an increase in sebaceous gland fate.

The work provides valuable genetics information on the mechanisms of HH action in skin and the data is generally convincing with quantification and statistical analysis.

For revisions, the data presented in the manuscript could be better organized into figures. A lot of valuable data regarding the Gli2/3/ dKO phenotype is hidden in the supplement and should be moved to the main figures, if possible. Have the authors considered a genomic approach such as scRNAseq to better understand the kind of changes in gene expression and cell identity induced by the some of the key mutants in this study. This kind of major revision if feasible to do, could really strengthen the depth of understanding for the current study. Finally, a thorough read of the manuscript with re-checking all the referenced panels is also needed. For example, there is no panel 3H in figure 3, but in text the authors write: "SOX9 was not significantly changed in Gli2/3EKO skin (Figure 3H)."

Referee #2:

In this paper, the authors report that epidermal cilia-dependent Gli2 regulates the subcompartments of the hair follicle (HF) stem cells and the sebaceous glands (SG). Additionally, the finding that Gli3 compensates for the function of Gli2 in the absence of Gli2 suggests that these transcription factors have redundant functions. Although the phenotypes observed in each genetically modified mouse model are clear, the interpretations of the specific functions of Gli2 and the compensatory function of Gli3 often involve speculative leaps and lack sufficient data. Below are specific points for consideration.

1. To better understand the dual role of Gli2 in cell fate determination and the redundant functions of Gli2 and Gli3, it would be necessary to present the expression patterns and gene expression levels of Shh, Gli1, Gli2, Gli3, Smo, and Ptc in the epidermis and hair follicles of both wild-type and Gli2-deficient mice. These data are essential for elucidating the functional interactions between Gli2 and Gli3.
2. In Figure 3C, E, and F, the hair follicle thickness appears to differ between control and Gli2EKO. Were these images captured at the same scale?
3. In the description of Figure 3E, an explanation is needed as to how "similarity" was interpreted. Moreover, without showing the temporal changes in the separation process, it is difficult to determine "similarity."
4. Quantitative data are also recommended for Figure 3F.
5. For Figure 4A and 4B, please provide quantitative results for the Gli3FL/Gli3R ratio. When focusing on the processing and functional changes of Gli3, it is appropriate to quantify the ratios of Gli3FL and Gli3R forms. In this case, verifying protein loading with housekeeping proteins and performing double normalization would enhance the reliability of the results.
6. The Gli3 transcription factor has dual functionality, acting as Gli3A (a transcriptional activator) and Gli3R (a transcriptional repressor). Would it not be important to examine Gli3A as well as Gli3R?
7. Regarding Supplementary Figure 1G, while the classification criteria for HF morphogenetic stages are referenced in the literature, it would be beneficial to briefly describe the criteria within this paper for the reader's convenience.
8. In Supplementary Figure 3D, E, G, and H, the detected genes are expressed not only in hair follicles but also in the epidermis. Preferably, lysates should be prepared exclusively from hair follicles to examine changes in gene expression levels. Additionally, p-values should be indicated in Figure 3G and H.
9. Regarding sample size, it is necessary to separately specify biological and technical replicates.

Minor points:

1. For readers with color vision deficiencies, the combination of red and green can be challenging to distinguish, so magenta and

green are preferable.

Point by point response

Referee #1:

The manuscript by Gozum et al represents a mouse genetics tour de force where the authors employ several single and double knockout mice to test the role and mechanisms of action of hedgehog (HH) signaling in skin epithelium, specifically in the formation of hair follicle and sebaceous gland. The authors find that Gli3 has overlapping roles with Gli2 that correspond with their function in the cilium. Gli3 also has a cilium independent function, as Gli12/3 dKO mice phenotype is not recapitulated in mutant mice lacking the cilium in the skin epithelium. When HH signaling is fully perturbed (such is the Gli2/3 dKO mice or Smo KO mice) the hair follicle morphogenesis is impaired, with multiple defects including: impaired hair follicle stem cell formation, impaired proliferation, lack of hair follicle differentiated lineages, adoption of what appears an epidermis like phenotype in the hair, with cyst-like structures taking the place of the hair follicle. This lack of hair lineages in the Gli2/3 mutants seems to be compensated by an increase in sebaceous gland fate.

The work provides valuable genetics information on the mechanisms of HH action in skin and the data is generally convincing with quantification and statistical analysis.

For revisions, the data presented in the manuscript could be better organized into figures. A lot of valuable data regarding the Gli2/3/ dKO phenotype is hidden in the supplement and should be moved to the main figures, if possible.

First of all, we are glad that the reviewer appreciates our extensive work analysing HH-cilia-Gli signaling during epidermal appendage formation.

We like to thank the reviewer for her/his supportive comments and valuable suggestions and have now rearranged the figures, e.g. we have moved the quantification of proliferation from Suppl. Fig. 1F to main Figure 1G and hair type analysis from Suppl. Fig. 1 G to main Figure 1H. Further, we are presenting the characterization of abnormal hair morphogenesis and differentiation of Gli2/3^{EKO} mice in main Figure 2 (prev. Suppl. Fig. 3).

Have the authors considered a genomic approach such as scRNAseq to better understand the kind of changes in gene expression and cell identity induced by the some of the key mutants in this study. This kind of major revision if feasible to do, could really strengthen the depth of understanding for the current study.

We addressed this helpful comment by performing a single-cell transcriptome analysis of Gli2^{EKO} and control mice at postnatal day (P) 2. Focusing on investigating the transcriptional changes in keratinocyte populations, our data reveal that distinct processes of HF cell lineage differentiation are abnormal and delayed in Gli2 mutants, thus supporting our initial findings. Further, the scRNAseq results show promotion of early SG differentiation in Gli2^{EKO} skin samples and are further strengthening our observation of a dual role for Gli2 in appendage formation. Importantly, the scRNAseq analyses helped to precisely pinpoint the affected keratinocyte populations in Gli2^{EKO} mice and the pathways involved. For example, Keratinocytes of cluster 13, encompassing HF matrix cells differentiating in hair lineages, showed a downregulation of cellular processes required for tissue growth, cellular rearrangement (Rho signaling and cytoskeletal organization) as well as

melanosome assembly in Gli2^{EKO} samples, most likely all reflecting the defect and delay in HF morphogenesis in Gli2^{EKO} mice. The novel scRNA seq data of P2 mouse skin keratinocytes from control and Gli2^{EKO} are shown in main Figure 5A-D, Figure EV4 and Figure EV5A-D. The original seq files have also been deposited at the GEO depository to be shared with the research community.

Finally, a thorough read of the manuscript with re-checking all the referenced panels is also needed. For example, there is no panel 3H in figure 3, but in text the authors write: "SOX9 was not significantly changed in Gli2/3EKO skin (Figure 3H)."

We apologize for this mistake. Most of the figures have been rearranged and we corrected the citation of figure panels, including the example pointed out by the reviewer.

Referee #2:

In this paper, the authors report that epidermal cilia-dependent Gli2 regulates the subcompartments of the hair follicle (HF) stem cells and the sebaceous glands (SG). Additionally, the finding that Gli3 compensates for the function of Gli2 in the absence of Gli2 suggests that these transcription factors have redundant functions. Although the phenotypes observed in each genetically modified mouse model are clear, the interpretations of the specific functions of Gli2 and the compensatory function of Gli3 often involve speculative leaps and lack sufficient data. Below are specific points for consideration.

The authors like to thank the reviewer for her/his comments and valuable suggestions, which helped to substantially improve the overall quality and depth of our study. We have revised the manuscript and our specific comments to the issues raised by the reviewer are outlined below.

1. To better understand the dual role of Gli2 in cell fate determination and the redundant functions of Gli2 and Gli3, it would be necessary to present the expression patterns and gene expression levels of Shh, Gli1, Gli2, Gli3, Smo, and Ptc in the epidermis and hair follicles of both wild-type and Gli2-deficient mice. These data are essential for elucidating the functional interactions between Gli2 and Gli3.

To address this comment, we have analysed the expression of HH components in Gli2^{EKO} and control skin by using single-cell RNAseq data at postnatal day (P) 2 generated during the revision process. The results are presented as feature plots in new main Figure 5C (Ptch1 and Gli1) and new Figure EV5A (Shh, Smo, Gli2 and Gli3). Briefly, Ptch1 and Gli1 were preferentially expressed in HF matrix/progenitor cells, and in ORS cells of clusters 11 and 5. Smo is also expressed by HF matrix cells and by ORS cells of cluster 11, whereas Shh expression is only detected in HF matrix/progenitor cells. Expression of both, Gli2 and Gli3 is detected in ORS cells and keratinocytes of the hair matrix and hair lineages, with strong expression of Gli3 in differentiating matrix cells, supporting our hypothesis that Gli3 could compensate for Gli2 in Gli2^{EKO} mice. All HH component were expressed by significantly fewer proportion of Gli2^{EKO} keratinocytes compared to control cells, likely due to the delay in HF development phenotype.

2. In Figure 3C, E, and F, the hair follicle thickness appears to differ between control and Gli2EKO. Were these images captured at the same scale?

The images were captured at the same scale. To demonstrate the difference between the HF size in epidermal tail whole mounts we have quantified thickness and length of the central follicle in HF triplets. The data show a significant increase in thickness and significant shortening of HFs in Gli2^{EKO} mice. These results are presented in new Figure 4C and 4D.

3. In the description of Figure 3E, an explanation is needed as to how "similarity" was interpreted. Moreover, without showing the temporal changes in the separation process, it is difficult to determine "similarity."

Given that the main focus of the manuscript is on the role of Gli signaling in HF and SG formation during early appendage formation, we have eliminated this sentence from the results.

4. Quantitative data are also recommended for Figure 3F.

BrdU+ve cells were quantified in Lrig1+ve progenitor cells in epidermal whole mounts from tail skin of P0 Gli2^{EKO} and control mice. The data show a significant increase in BrdU+ve cells in Gli2^{EKO} epidermis and this result is presented in new Figure 4F.

5. For Figure 4A and 4B, please provide quantitative results for the Gli3^{FL}/Gli3^R ratio. When focusing on the processing and functional changes of Gli3, it is appropriate to quantify the ratios of Gli3^{FL} and Gli3^R forms. In this case, verifying protein loading with housekeeping proteins and performing double normalization would enhance the reliability of the results.

We have now repeated the western blot and provide the quantification for Gli3 processing in Gli2^{EKO} and control mice. As shown in Figure 6A and 6B there is significantly less of Gli3 repressor to activator forms in Gli2^{EKO} mice supporting our hypothesis that Gli3 can function as an activator to compensate for Gli2 loss in Gli2^{EKO} animals.

6. The Gli3 transcription factor has dual functionality, acting as Gli3A (a transcriptional activator) and Gli3R (a transcriptional repressor). Would it not be important to examine Gli3A as well as Gli3R?

We have addressed this issue by quantifying the full-length Gli3 protein in western blots presented in Figure 6A. Full-length Gli3 (Gli3^{FL}, activator) is significantly increased in Gli3^{FL} : Gli3^R ratio in Gli2^{EKO} mice compared to control littermates (please see also response to 5.).

7. Regarding Supplementary Figure 1G, while the classification criteria for HF morphogenetic stages are referenced in the literature, it would be beneficial to briefly describe the criteria within this paper for the reader's convenience.

We have included some of the main criteria for HF morphogenesis stage 4 and stage 6 in the manuscript text in the Material and Method sections, chapter "histological analyses": "According to Saxena et al, 2018, the following criteria applied for stage 4: The pre-mature hair placode is invaginating into dermis with the dermal condensate closely associated to the growing tip of the placode. At stage 6, when the HF looks more mature, the hair matrix is beginning to surround the dermal condensate and a hair shaft starts to appear."

8. In Supplementary Figure 3D, E, G, and H, the detected genes are expressed not only in hair follicles but also in the epidermis. Preferably, lysates should be prepared exclusively from hair follicles to examine changes in gene expression levels. Additionally, p-values should be indicated in Figure 3G and H.

To address the reviewer's comment, we are now providing additional quantifications for marker expression. Unfortunately, hair follicles cannot easily be isolated from abnormal skin tissue of Gli2/3^{EKO} mice, due to cyst-like structures growing deep in the dermal tissue. However, we performed extensive stainings for K10, filaggrin, K15 and CD34 for quantifications that were performed as follows:

Quantification of K10 and Filaggrin:

Fluorescence images of the back skin sections from P6 and P49 control and Gli2/3^{EKO} mice were analysed using custom Fiji macros. The intensity was quantified using a macro to process files containing regions of interest (ROIs), including background subtraction. The ROIs corresponded to the area between the HF infundibulum and the distal tip of the HF within the skin tissue. To exclude potential artifacts, a mask was generated by thresholding the images to remove pixels with intensity values exceeding 8000 arbitrary units (A.U.) for K10 and at 11000 A.U. for Filaggrin. The filtered images were then quantified within the ROIs to extract mean fluorescence intensity.

Quantification of K15 and CD34:

Fluorescence images of the back skin sections from P6 and P49 control and Gli2/3^{EKO} mice were examined for the presence of K15 or CD34 in HF structures and HF were classified as either positive or negative. Only regions of HFs where K15 or CD34 expression is typically expressed were analysed.

In addition, qRT-PCR experiments were performed for Krt10, filaggrin, Krt15 and CD34 expression from epidermis isolated from P6 and P49 control and Gli2/3^{EKO} mice.

These data are now presented in new Figure 2C-K for P6 mice and in new Figure EV3A-L for P49 adult mice.

We have added p-values for all quantifications.

9. Regarding sample size, it is necessary to separately specify biological and technical replicates.

We included this important information in the figure legends for the individual experiments presented.

Minor points:

1. For readers with color vision deficiencies, the combination of red and green can be challenging to distinguish, so magenta and green are preferable.

We like to thank the reviewer for the important suggestion and have now used green and grey to display double staining imaging results, e.g. in Figure 1F, 2C,F and Figure 4G and Figure EV3A,D.

Dear Catherin,

Thank you for submitting your revised manuscript (EMBOJ-2024-119233R) to The EMBO Journal, as well for your patience with our response. Your amended study was sent back to the two referees for their scientific reassessment, and we have received detailed re-reports from both, which I enclose below. As you will see, the experts state that the work has been substantially enhanced by the revisions and they are now broadly in favour of publication.

Thus, we are pleased to inform you that your manuscript has been accepted in principle for publication in The EMBO Journal.

We now need you to take care of a number of minor issues related to formatting and data annotation as detailed below, which need to be addressed at resubmission of the work.

Please contact me at any time if you have additional questions related.

As you might have noted from our webpage, every paper at the EMBO Journal now includes a 'Synopsis', displayed on the html and freely accessible to all readers. The synopsis includes a 'model' figure as well as 2-5 one-short-sentence bullet points that summarize the article. I would appreciate if you could provide this figure. Also, please remove the bullet points from the manuscript and provide it in a separate text file.

Thank you for giving us the chance to consider your manuscript for The EMBO Journal. I look forward to your final revision.

Again, please contact me at any time if you need any help or have further questions.

Best regards,

Daniel

>> Author information: please revisit e-mail contact provided in our system for co-author P.H.; revisit the name information consistency for G.G. .

>> Limit the keywords for your study to maximally five.

>> Author Contributions: Remove the author contributions information from the manuscript text. Note that CRediT has replaced the traditional author contributions section as of now because it offers a systematic machine-readable author contributions format that allows for more effective research assessment. and use the free text boxes beneath each contributing author's name to add specific details on the author's contribution.

More information is available in our guide to authors.
<https://www.embopress.org/page/journal/14602075/authorguide>

>> Dataset EV legends: Table EV1 should be renamed Table 1.

>> Figure callouts: callouts for Table EV1 and Suppl. Table should be corrected to Table 1.

>> Please recheck the reference for the bioRxiv entry Damen et al. (2021) and update the citation if in the meantime published as regular article.

>> Funding: please enter all funding information in the list of funders in our online system: "e.g 40749390, 423957469, 73111208 etc...".

>> Remove the Reagents and Tools table from the manuscript and provide as a separate file using the existing template in the Guide For Authors, listing key reagents, experimental models, software and relevant equipment.

>> Data availability section: please remove the referee token and make sure privacy is released from the GEO dataset.

>> Please indicate redisplay of data from Fig.2A in the figure legend of Figure EV2A.

>> Consider additional changes and comments from our production team as indicated below:

- Figure legends:

1. Please note that the exact p values are not provided in the legends of figures 1C, G, H; 2B, D, J; 3F, G; 4B, F, H; EV3 E, H, K.
2. Please indicate the statistical test used for data analysis in the legends of figures EV5 C, D.
3. Please note that information related to n is missing in the legend of figure 6B
4. Please note that n=2 in figure 6C
5. Please note that the error bars are not defined in the legends of figures 6B, C

Referee #1:

The authors addressed my original concerns, including substantial additional data in the form of scRNA-seq. I believe the manuscript is now ready for publication.

Referee #2:

I would like to thank the authors for thoroughly addressing my previous comments. In particular, I appreciate the additional quantification of the data, which clearly strengthens the conclusions. The manuscript has improved overall, and most of the concerns I raised in the first review have been adequately resolved. In my opinion, the manuscript now meets all the criteria for publication.

The authors addressed the remaining editorial issues.

Dear Catherin,

Thank you for submitting the revised version of your manuscript. I have now evaluated your amended manuscript and concluded that the remaining minor concerns have been sufficiently addressed.

I am thus pleased to inform you that your manuscript has been accepted for publication in the EMBO Journal.

On a different note, I would like to alert you that EMBO Press offers a format for a video-synopsis of work published with us, which essentially is a short, author-generated film explaining the core findings in hand drawings, and, as we believe, can be very useful to increase visibility of the work. Please see the following link for representative examples and their integration into the article web page:

<https://www.embopress.org/doi/full/10.15252/embj.2019103932>

Best regards,

Daniel

Daniel Klimmeck, PhD
Senior Editor
The EMBO Journal
EMBO
Postfach 1022-40
Meyerhofstrasse 1
D-69117 Heidelberg
contact@embojournal.org